



# Assessment of the southern polar and subpolar warming in the PMIP4 Last Interglacial simulations using paleoclimate data syntheses

Qinggang Gao[1, 2], Emilie Capron[3], Louise C. Sime[1], Rachael H. Rhodes[2], Rahul Sivankutty[1], Xu Zhang[1], Bette L. Otto-Bliesner[4], and Martin Werner[5]

[1]Ice Dynamics and Paleoclimate, British Antarctic Survey, Cambridge, United Kingdom
[2]Department of Earth Sciences, University of Cambridge, Cambridge, United Kingdom
[3]Université Grenoble Alpes, CNRS, IRD, Grenoble INP, IGE, Grenoble, France
[4]Climate and Global Dynamics Laboratory, NSF National Center for Atmospheric Research, Boulder, CO, USA
[5]Alfred Wegener Institute, Helmholtz Centre for Polar and Marine Research, Bremerhaven, Germany

**Correspondence:** Qinggang Gao (qino@bas.ac.uk)

**Abstract.** Given relatively abundant paleo proxies, the study of the Last Interglacial (LIG, ∼129-116 thousand years ago, ka) is valuable to understanding natural variability and feedback in a warmer-than-preindustrial climate. The Paleoclimate Modelling Intercomparison Project Phase 4 (PMIP4) coordinated LIG model simulations which focus on 127 ka. Here we evaluate 12 PMIP4 127-ka Tier 1 model simulations against four recent paleoclimate syntheses of LIG sea and air temperatures and sea ice concentrations. The four syntheses include 99 reconstructions and show considerable variations, some but not all of which are attributable to the different sites included in each synthesis. All syntheses support the presence of a warmer Southern Ocean, with reduced sea ice, and a warmer Antarctica at 127 ka compared to the preindustrial. The PMIP4 127-ka Tier 1 simulations, forced solely by orbital parameters and greenhouse gas concentrations, do not capture the magnitude of this warming. Here we follow up on previous work that suggests the importance of preceding deglaciation meltwater release into the North Atlantic. We run a 3000-year 128-ka simulation using HadCM3 with a 0.25 $Sv$ North Atlantic freshwater hosing, which approximates the PMIP4 127-ka Tier 2 H11 (Heinrich event 11) simulation. The hosed 128-ka HadCM3 simulation captures much of the warming and sea ice loss shown in the four data syntheses at 127 ka relative to preindustrial: south of 40° S, modelled annual sea surface temperature (SST) rises by 1.3±0.6°C, while reconstructed average anomalies range from 2.2°C to 2.7°C; modelled summer SST increases by 1.1±0.7°C, close to 1.2-2.2°C reconstructed average anomalies; September sea ice area (SIA) reduces by 40%, similar to reconstructed 40% reduction of sea ice concentration (SIC); over the Antarctic ice sheet, modelled annual surface air temperature (SAT) increases by 2.6±0.4°C, even larger than reconstructed average anomalies 2.2°C. Our results suggest that the impacts of deglaciation ice sheet meltwater need to be considered to simulate the Southern Ocean and Antarctic changes at 127 ka.



## 1 Introduction

The LIG appears as one of the warmest interglacials in paleoclimate records over the past 800 thousand years (kyr, Past Interglacials Working Group of PAGES, 2016). Large parts of the globe experienced a warmer-than-preindustrial climate during the LIG (*e.g.* CAPE-Last Interglacial Project Members, 2006; Hoffman et al., 2017; Fischer et al., 2018). The Arctic experienced a summer warming of 3.7±1.5 K at 127 ka compared to the preindustrial (Sime et al., 2023), which is comparable to projected regional warming for the end of this century (IPCC, 2021). Whilst the warm polar climate during the LIG resulted

from orbital forcings, unlike current anthropogenic global warming mainly caused by greenhouse gas emissions (Otto-Bliesner et al., 2017), the internal climate feedbacks such as sea ice feedbacks are akin to those of the present day (Diamond et al., 2021, 2023). Therefore, the LIG provides unique insights to improve our understanding of climate feedback processes in a warmer-than-preindustrial climate (Thomas et al., 2020).

Global and regional surface temperature estimates during the LIG have been inferred from natural climate archives. Com-

piling 263 surface temperature reconstructions from terrestrial, marine, and ice core records, Turney and Jones (2010) were the first to propose a global mean maximum LIG warming of approximately 1.9°C above preindustrial levels. However, this study has several limitations. First, the paleoclimate records were taken on their original age scales, which can introduce absolute dating uncertainties of up to 6 kyr (see Govin et al., 2015, for a review). Second, this dataset compiles peak warmth values during the LIG at each site, hence assuming a synchronous LIG global maximum warming. However, numerous studies demonstrated

that LIG surface temperature peaks did not occur synchronously across the globe (*e.g.* Cortijo et al., 1999; Bauch et al., 2011; Govin et al., 2012). Based on SST reconstructions with a consistent chronology, and accounting for uncertainties attached to dating and SST reconstruction methods, Hoffman et al. (2017) estimated a global annual SST warmth of 0.5±0.3°C at 125 ka above preindustrial levels. Assuming a scaling factor of 1.6 from global mean SST to SAT (Snyder, 2016), global annual SAT could be ~0.8±0.5°C warmer than preindustrial levels (Fischer et al., 2018). Hoffman et al. (2017) also found that the

magnitude of warming was greater in polar and subpolar regions, as shown previously by Capron et al. (2014, 2017). Chandler and Langebroek (2021a) also estimated a Southern Ocean annual SST anomaly of 1.6±1.1°C at 126 ka relative to present.

Antarctic sea ice regulates the transfer of mass, momentum, and energy between the atmosphere and the ocean. While the short period of instrumental records might not be enough to constrain variability and impacts of Antarctic sea ice (Rackow et al., 2022), relevant climate processes can be inferred from paleo proxies (Crosta et al., 2021). Past Antarctic sea ice changes have

been reconstructed from diatom fossil assemblages in marine sediment cores and sea-salt sodium in Antarctic ice cores (Crosta et al., 2022; Wolff et al., 2006). Based on new SIC and SST reconstructions from nine marine sediment cores, Chadwick et al. (2023) suggested that the reduced sea ice cover over the Southern Ocean at 127 ka compared to the preindustrial may be related to the H11 meltwater from the Northern Hemisphere. This makes sense since the warmer Southern Ocean and reduced sea ice from this time cannot be directly attributed to solar forcings, because austral summer solar insolation was lower than present

at 127 ka (Berger and Loutre, 1991). It is also not attributable to radiative forcings of greenhouse gases, because atmospheric concentrations of carbon dioxide, methane, and nitrous oxide were all lower than preindustrial levels (Otto-Bliesner et al., 2017). Whilst other environmental conditions, such as the volume and configuration of the Antarctic ice sheet (Golledge et al.,



2021), are poorly constrained, previous work suggests that a reduced Antarctic ice sheet might not be responsible for the Southern Ocean warming (Holloway et al., 2016).

Recent modelling efforts have been undertaken to address this knowledge gap. Analysis of PMIP Phase 3 (PMIP3) LIG simulations and the synthesis from Turney and Jones (2010) indicated that all models underestimated the magnitude of climate anomalies suggested by the proxies (Lunt et al., 2013). However, firm conclusions could not be drawn because of inconsistent experiment protocols and the limitations associated with the use of a LIG peak-warmth-centred synthesis. Subsequently, the PMIP4 project provided consistent boundary conditions for 127 ka, known as the *lig127k* experiment (Otto-Bliesner et al.,

2017). Otto-Bliesner et al. (2021) compared surface temperature from a *lig127k* simulation ensemble with multiple data syntheses. They showed that the ensemble of 17 different models did not reproduce the pronounced 127-ka warmth at southern mid-to-high latitudes. However, they did not investigate the output from individual *lig127k* simulations, nor did they explore mechanisms responsible for the model-data mismatch.

Many studies suggested that freshwater input from northern ice sheets into the North Atlantic during the penultimate
deglaciation likely warmed southern mid-to-high latitudes during the early LIG (Govin et al., 2012; Capron et al., 2014; Stone et al., 2016; Holloway et al., 2018; Menviel et al., 2019; Chadwick et al., 2023). Otto-Bliesner et al. (2017) therefore included a protocol for a Tier 2 sensitivity experiment, the *lig127k-H11*, alongside the primary Tier 1 *lig127k* experiment. The *lig127k-H11* protocol requires the addition of a meltwater flux of 0.2 Sv (1 Sverdrup $= 10^6 \ m^3 s^{-1}$) to the North Atlantic between 50° N and 70° N for 1000 years. Guarino et al. (2023) used HadGEM3 to run 250 years of the *lig127k-H11* simulation,
which is too short to show significant changes in the Southern Ocean. Holloway et al. (2018) suggested that 3000 years of H11 hosing is likely required to capture the full magnitude of the Southern Ocean and Antarctic warming.

The manuscript is organised as below: Section 2 introduces data and methods, including a new 3000-year 128-ka-H11 simulation using HadCM3; Section 3 presents results; discussions are presented in Section 4 and conclusions in Section 5.

## 2   Materials

### 2.1   PMIP4 Tier 1 *lig127k* model simulations

PMIP4 *lig127k* simulations used orbital parameters and atmospheric greenhouse gas concentrations obtained from ice core records at 127 ka (Otto-Bliesner et al., 2017). We examine climate anomalies in 12 *lig127k* simulations relative to corresponding preindustrial control simulations (*piControl*, Eyring et al., 2016). Configurations of the models are presented in Table 1. CNRM-CM6-1 used the same greenhouse gas concentrations in *lig127k* as in *piControl*. We construct a PMIP4 model ensem-
ble from the 12 models, and for comparison, we also use the PMIP3 model ensemble of 10 models presented in Lunt et al. (2013).

For data analysis, we use 100-year simulation from the end of each model integration period. All model outputs are interpolated onto 1°×1° grids using bilinear interpolation before analysis. To extract model output at each site, nearest neighbour interpolation is used. Summer SST is defined as the average January-February-March SST. SIA is estimated by summing the
product of Antarctic SIC and grid cell area.



**Table 1.** Configurations of models that carried out PMIP4 *lig127k* simulations. The table is adapted from Kageyama et al. (2021).

| Model name | Model components: Atmosphere; Land; Ocean; Sea Ice | Model grid (i_lon × i_lat × z_lev): Atmosphere; Ocean | Boundary conditions (if not prescribed): Vegetation; Aerosol; Ice-Sheet | Spin-up length (years) | References |
|---|---|---|---|---|---|
| ACCESS-ESM1-5 | UM; CABLE2.4; MOM5; CICE4.1 | 192×145×L38; 360×300×L50 | - | 500 | Ziehn et al. (2017) |
| AWI-ESM-1-1-LR | ECHAM6.3.04p1; JSBACH_3.20 FESOM 1.4; FESOM 1.4 | 192×96×L47; 126,859 nodes × L46 | Vegetation: Interactive | 1000 | Sidorenko et al. (2015) |
| CESM2 | CAM6; CLM5; POP2; CICE5.1 | 288×192×L32; 320×384×L60 | Aerosol: interactive | 925 | Otto-Bliesner et al. (2020) |
| CNRM-CM6-1 | ARPEGE-Climat; ISBA-CTRIP; NEMO3.6; GELATO6 | 256×128×L91; 362×294×L75 | - | 200 | Voldoire et al. (2019) |
| EC-Earth3-LR | IFS-cy36r4; HTESSEL; NEMO3.6; LIM3 | 480×240×L62; 362×292×L75 | - | 400 | Hazeleger et al. (2012) |
| FGOALS-g3 | GAMIL3; CLM4.5; LICOM3; CICE4 | 180×90×L26; 36×218×L30 | - | 900 | Zheng et al. (2020) |
| GISS-E2-1-G | GISSE2.1; GISSE2.1; GISS Ocean v1; GISS Ocean v1 | 144×90×L40; 360×180×L40 | Aerosol: NINT | 1000 | Kelley et al. (2020) |
| HadGEM3-GC31-LL | MetUM-GA7.1; JULES-GA7.1; NEMO-GO6.0; CICE-GSI8 | 192×144×L85; 360 × 330 × L75 | - | 450 | Williams et al. (2018) |
| IPSL-CM6A-LR | LMDZ6; ORCHIDEE; NEMO-OPA; NEMO-LIM3 | 144 × 143 × L79; 362 × 332 × L75 | Vegetation: prescribed PFTs, interactive phenology | 400 | Boucher et al. (2020) |
| MIROC-ES2L | CCSR AGCM; MATSIRO6.0 + VISIT-e; COCO4.9; COCO4.9 | 128 × 64 × L40; 360 × 256 × L63 | - | 1450 | O'Ishi et al. (2021) |
| NESM3 | ECHAM6.3; JS-BACH; NEMO3.4; CICE4.1 | 192×96×L47; 384 × 362 × L46 | Vegetation: interactive | 500 | Cao et al. (2018) |
| NorESM2-LM | CAM-OSLO; CLM; BLOM; CICE | 144×96×L32; 360 × 384 × L53 | - | 400 | Zhang et al. (2019) |



## 2.2 LIG climate data syntheses

In this study, we use the four most recent surface temperature data syntheses and one SIC dataset.

First, we use surface temperature reconstructions at 127 ka distributed over southern polar and subpolar regions from Capron et al. (2017). In total, we consider two annual SST records, 15 summer SST records, and four annual SAT records. The SST records were derived from marine sediment cores using foraminiferal Mg/Ca ratios, alkenone unsaturation ratios, and micro-fossil faunal assemblage transfer functions (including diatoms, radiolarians, and foraminifera; Capron et al., 2014). Uncertainties in the SST reconstructions might come from advection/dispersion, seasonality, and non-thermal influences (Chandler and Langebroek, 2021b). The SAT records were deduced from water isotope profiles in four Antarctic ice cores (Masson-Delmotte et al., 2011), using present-day spatial relationships between water isotopic compositions of Antarctic snow and surface temperature (Jouzel et al., 2007). There are some uncertainties in the approximation of the observed spatial relationships as temporal relationships across glacial-interglacial timescales (*e.g.* Werner et al., 2018; Masson-Delmotte et al., 2011; Sime et al., 2009). It was shown that past temperature estimates based on this method could be underestimated for warmer-than-present-day conditions (Sime et al., 2009). The paleo records have a minimum temporal resolution of 2 kyr and they have all been transferred on the AICC2012 ice core chronology using climate alignment hypotheses (details provided in Capron et al., 2014). A Monte Carlo approach was used to determine temperature reconstructions for the 126-128 ka interval, referred to as a 127 ka time slice, and associated uncertainties attached to both dating and reconstruction methods. The resulting average uncertainty ($2\sigma$) is about 2.6°C and 3.0°C for SST and SAT, respectively. Capron et al. (2017) provided 127 ka surface temperature anomalies relative to the HadISST1 dataset (1870-1899, Rayner et al., 2003).

Second, we use SST reconstructions from Hoffman et al. (2017). Here we consider 12 annual SST records and seven summer SST records over the Southern Ocean. These SST reconstructions were based on foraminiferal Mg/Ca ratios, alkenone unsaturation ratios, and microfossil faunal assemblage transfer functions (including diatoms, radiolarians, foraminifera, and coccoliths). Annual SST from microfossil assemblages was calculated as an average of "August summer" and "February winter" SST. These records have a minimum temporal resolution of 4 kyr and were placed in a common temporal framework using the Speleo-Age model (Barker et al., 2011). Hoffman et al. (2017) compiled LIG SST anomalies relative to HadISST1 (1870-1889). The 127 ka surface temperature time slice and associated uncertainties were extracted by Capron et al. (2017) following Hoffman et al. (2017). The age difference between this age model and AICC2012 is about 1.4 kyr at 127 ka (Capron et al., 2017).

The third recent Southern Ocean SST synthesis was compiled by Chandler and Langebroek (2021a). There are 20 annual SST reconstructions and 21 summer SST reconstructions. They used two geochemical proxies (alkenones and *Globigerina bulloides* Mg/Ca ratios) and two assemblage-based proxies (foraminifera and diatoms), with consistent calibration functions. Age models of the records were established through benthic $\delta^{18}O$ record alignments on the LS16 chronology (Lisiecki and Stern, 2016). This compilation extends from the present to the penultimate glacial period with a 2 kyr interval. Because of discontinuous SST records, we cannot extract a 127 ka value for each site based on linear interpolation; hence, we use their reconstructed values at 128 ka. Although Chandler and Langebroek (2021a) provided calibration errors related to each



proxy, they did not estimate uncertainties accounting for both chronological and reconstruction errors. As they only provided temperature anomalies relative to the World Ocean Atlas 2018, we estimated temperature anomalies relative to the HadISST1 dataset (1870-1899).

    Lastly, Chadwick et al. (2021) retrieved both September SIC and summer SST from nine marine cores between 50 and 70° S across the LIG. The Chadwick datasets focus on a more southerly region compared with the previous synthesis, since they were

generated with the express purpose of looking at the suggested halving of winter-time Antarctic sea ice extent. The positions of the cores in this dataset are thus more aligned with the poleward LIG wintertime sea ice edge (Holloway et al., 2017; Chadwick et al., 2023). Chadwick et al. (2021) used diatom assemblage transfer functions based on a modern analogue technique (Crosta et al., 1998). Their data were presented on the AICC2012 age model. Reconstruction errors were estimated as 1.09°C for SST and 9% for SIC, and the dating uncertainties are about 2.6 kyr (Chadwick et al., 2022). As the reconstructions were published

as time series on irregular time steps, we used linear interpolation to extract values at 127 ka. We estimated September SIC and summer SST anomalies relative to the HadISST1 dataset (1870-1899).

    In total, we gathered 34 annual SST reconstructions, 52 summer SST reconstructions, nine September SIC reconstructions, and four annual SAT reconstructions (Fig. 1). Among these 99 records, there are 20 pairs of reconstructions based on the same marine sediment cores in at least two different datasets (Table A1). SST estimates from Chandler and Langebroek (2021a)

always lie within the uncertainty range of those in Capron et al. (2017), whereas SST estimates in Hoffman et al. (2017) can deviate significantly from those in Capron et al. (2017, e.g. core MD94-102) or Chandler and Langebroek (2021a, e.g. core MD97-2121). In addition to the fact that different proxies have been used for some reconstructions, these discrepancies likely result from different dating, calibration, and resampling techniques (Capron et al., 2017). Due to different reference chronologies and approaches for estimating the 127 ka data and uncertainties, these four datasets are treated as independent

data benchmarks.

    At the regional scale, all syntheses suggest notably positive climate anomalies at 127 ka relative to preindustrial levels over southern polar and subpolar regions (Table 2). The Hoffman et al. (2017) dataset indicates larger anomalies of annual SST than summer SST (2.7±1.0 vs. 1.6±0.9°C) using different marine cores, whereas the Chandler and Langebroek (2021a) dataset shows similar magnitude of anomalies (2.2±1.5 vs. 2.2±1.9°C). The Chadwick et al. (2021) dataset suggests the smallest

regional summer SST anomaly (1.2±1.1°C), which might result from the more southerly site locations (Fig. 1b).





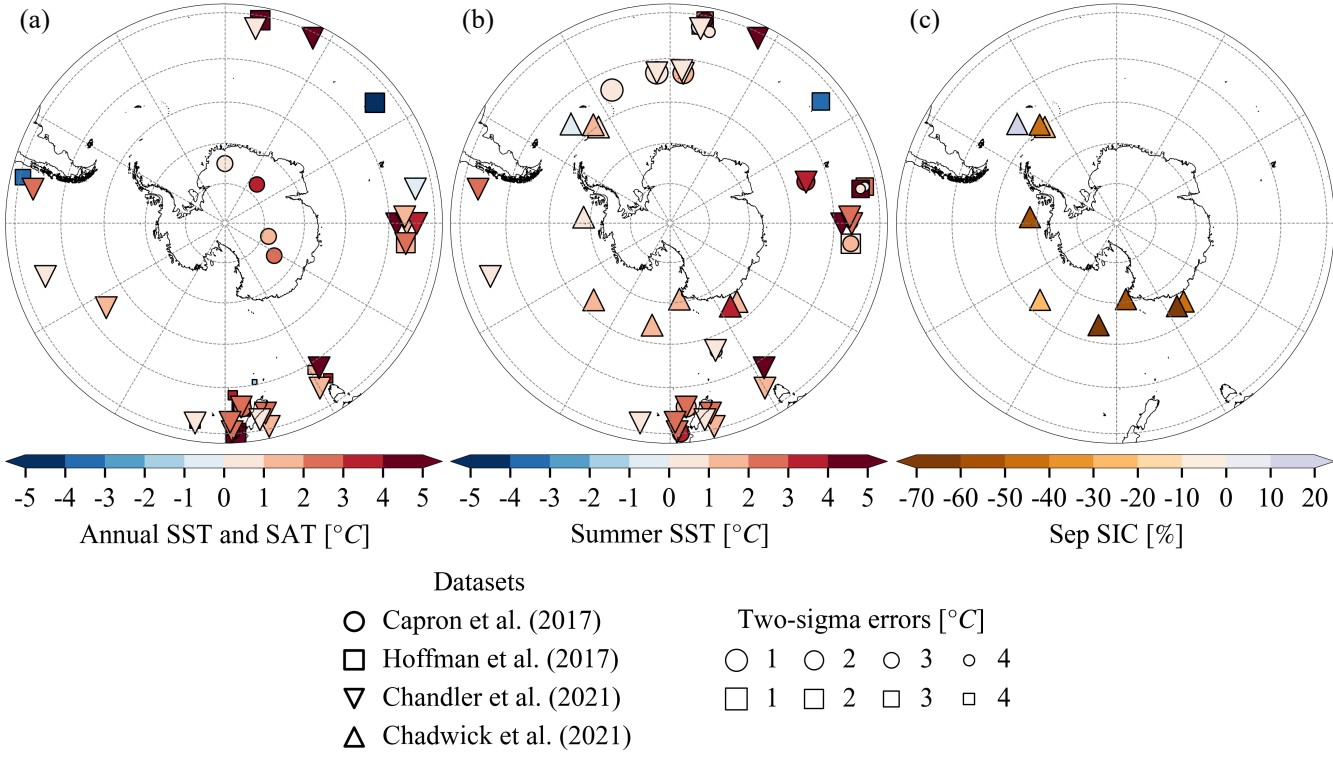

**Figure 1.** Reconstructed climate anomalies at 127 ka relative to preindustrial: (a) annual SST and SAT; (b) summer SST; and (c) September SIC. For Capron et al. (2017) and Hoffman et al. (2017) syntheses, the sizes of the symbols denote two-sigma errors associated with reconstructions.

**Table 2.** Regional average surface temperature and SIC anomalies at 127 ka relative to preindustrial from the four climate syntheses presented in Fig. 1. Values for Capron et al. (2017) and Hoffman et al. (2017) are extracted directly from Capron et al. (2017), where they estimated area weighted average anomalies and associated two-sigma errors. For Chandler and Langebroek (2021a) and Chadwick et al. (2021) datasets, we calculated mean values and one standard deviation directly from all records. The number of cores used for calculation is given in parentheses.

| Syntheses | Annual SST [°C] | Summer SST [°C] | Annual SAT [°C] | September SIC [%] |
|---|---|---|---|---|
| Capron et al. (2017) | - | $1.8 \pm 0.8$ (15) | $2.2 \pm 1.4$ (4) | - |
| Hoffman et al. (2017) | $2.7 \pm 1.0$ (12) | $1.6 \pm 0.9$ (7) | - | - |
| Chandler and Langebroek (2021a) | $2.2 \pm 1.5$ (20) | $2.2 \pm 1.9$ (21) | - | - |
| Chadwick et al. (2021) | - | $1.2 \pm 1.1$ (9) | - | $-41.2 \pm 24.9$ (9) |



## 2.3 HadCM3 simulations

We use the Hadley Centre Climate model HadCM3 to investigate LIG climate responses to meltwater forcing over the North
Atlantic. The HadCM3 couples a hydrostatic atmospheric component HadAM3 and a barotropic ocean component HadOM3
(Tindall et al., 2009). The HadAM3 has a horizontal resolution of 2.5°×3.75° and 19 hybrid vertical levels, while the HadOM3
has a horizontal resolution of 1.25°×1.25° and 20 oceanic levels. Using HadCM3, Holloway et al. (2018) explored the effects
of northern ice sheet meltwater associated with H11 on the 128 ka Antarctic climate. They applied a uniform freshwater forcing
of 0.25 $Sv$ over the North Atlantic between 45° N and 70° N in a 1600-year simulation. While the southern polar and subpolar
warming signals were partially reproduced in the simulation, they suggested that a longer simulation might reconcile the
paleoclimate proxies. Although HadCM3 was not used for PMIP4 model simulations, it is unfortunately not computationally
feasible to run more recent CMIP6 UK models for more than a few hundred years (Guarino et al., 2023), making extending
the Holloway et al. (2018) H11 simulation the only feasible option with a UK CMIP model. As recommended by Holloway
et al. (2018), we extend this freshwater-forced 128-ka simulation to 3000 years (referred to as HadCM3_128k_H11 hereafter).
For comparison, we also run a standard 127-ka simulation of HadCM3 (HadCM3_127k), which gives qualitatively consistent
results with a standard 128-ka simulation of HadCM3 by Holloway et al. (2018, not shown). Climate anomalies are estimated
with respect to a preindustrial simulation of HadCM3. Analysis of the HadCM3 simulations follows that of PMIP4 *lig127k*
simulations.

## 2.4 Other materials

The HadISST1 dataset is constructed by the UK Met Office using multiple observational data sources (Rayner et al., 2003).
It contains global monthly SST and SIC on 1°×1° grids from 1870 to present. While the quality of HadISST1 is afflicted by
sparse observations during 1870-1899 over the Southern Ocean, we use it to highlight potential model bias in the preindustrial
simulations.

## 2.5 Methods

Here we introduce the methods used for model-data comparisons.

Root mean squared errors (RMSE) between gridded datasets are area-weighted as

$$RMSE = \sqrt{\frac{\sum_i (A_i \times (X1_i - X2_i)^2)}{\sum_i A_i}}, \tag{1}$$

where $i$ refers to a grid cell, $A$ grid cell area, $X1$ and $X2$ the same variable in the first and second datasets, respectively.

RMSE between a gridded dataset and a collection of site values are defined as

$$RMSE = \sqrt{\frac{\sum_m^N (X3_m - X4_m)^2}{N}}, \tag{2}$$

where $m$ refers to a site, $N$ the number of sites, $X3$ values extracted from the gridded dataset using nearest neighbour interpo-
lation for each site $m$, and $X4$ site values.





## 3    Results

### 3.1    Preindustrial simulations vs. the HadISST1 dataset

Figure 2 compares annual SST in the 12 *piControl* simulations with HadISST1 over the Southern Ocean. The RMSE between each simulation and HadISST1 ranges from 0.8°C (IPSL-CM6A-LR) to 3.3°C (MIROC-ES2L), and the RMSE between the PMIP4 model ensemble and HadISST1 is 1.1°C (Table 3). The best agreement with HadISST1 is achieved by CNRM-CM6-1 and IPSL-CM6A-LR, with RMSE less than 1°C (Fig. 2d and 2i). The largest bias compared to HadISST1 is found in EC-Earth3-LR and MIROC-ES2L (Fig. 2e and 2j; RMSE: 2.7 and 3.3°C). While spatial patterns of the bias are different across models, all models except FGOALS-g3 exhibit notable warm bias over the southern Indian Ocean. In contrast, four models exhibit widespread cold bias over the southern Pacific Ocean: AWI-ESM-1-1-LR, CESM2, FGOALS-g3, and NESM3 (Fig. 2b, 2c, 2f, and 2k).

For summer SST, the magnitude of RMSE between simulations and HadISST1 is generally larger than that for annual SST (Table 3). The RMSE varies between 1.0°C (HadGEM3-GC31-LL) and 3.8°C (MIROC-ES2L), and reaches 1.4°C for the PMIP4 model ensemble. Here FGOALS-g3 and HadGEM3-GC31-LL have the lowest RMSE (~1.1°C), whereas EC-Earth3-LR and MIROC-ES2L again show the largest RMSE (~3.8°C). Spatial patterns of the bias are similar to those in Fig. 2 (not shown).

Regarding September sea ice, relative differences in September SIA between simulations and HadISST1 have a large range: from -85% (MIROC-ES2L) to 6% (IPSL-CM6A-LR) for individual models and -17% for the PMIP4 model ensemble (Table 3). MIROC-ES2L simulates the smallest September SIA (85% less than HadISST1), followed by EC-Earth3-LR (44% less) and NorESM2-LM (40% less). AWI-ESM-1-1-LR and FGOALS-g3 demonstrate a good match to HadISST1 (within 3%) due to regional compensation of positive and negative bias (not shown).





**Figure 2.** Differences in annual SST between *piControl* simulations by the 12 models listed in Table 1 and HadISST1 (1870-1899). We only show differences that are significant at 5% level based on the student's t-test with Benjamini-Hochberg Procedure controlling false discovery rates (Benjamini and Hochberg, 1995).





**Table 3.** Comparison of *piControl* simulations against HadISST1 (1870-1899), and comparison of *lig127k* simulations against *piControl* simulations, for each model and two model ensembles. RMSE and mean differences are both area-weighted. One standard deviation is given after mean temperature differences and represents spatial variability.

| Models | *piControl* vs. HadISST1 (RMSE over the ocean south of 40° S) | | | *lig127k* vs. *piControl* (mean differences over the ocean south of 40° S) | | | |
|---|---|---|---|---|---|---|---|
| | Annual SST [°C] | Summer SST [°C] | Sep SIA$^a$ [%] | Annual SST [°C] | Summer SST [°C] | Annual SAT$^b$ [°C] | Sep SIA$^c$ [%] |
| ACCESS-ESM1-5 | 1.7 | 2.5 | -30 | $0.4 \pm 0.8$ | $0.2 \pm 1.0$ | $1.6 \pm 0.4$ | -34 |
| AWI-ESM-1-1-LR | 1.4 | 1.5 | 3 | $0.2 \pm 0.4$ | $-0.1 \pm 0.5$ | $0.5 \pm 0.4$ | -7 |
| CESM2 | 1.3 | 1.3 | -23 | $0.1 \pm 0.4$ | $0.0 \pm 0.6$ | $0.2 \pm 0.2$ | -4 |
| CNRM-CM6-1 | 0.9 | 1.4 | -7 | $0.2 \pm 0.6$ | $-0.1 \pm 0.7$ | $0.7 \pm 0.3$ | -9 |
| EC-Earth3-LR | 2.7 | 3.7 | -44 | $-0.2 \pm 0.3$ | $-0.7 \pm 0.4$ | $-0.1 \pm 0.2$ | -4 |
| FGOALS-g3 | 1.1 | 1.1 | 0 | $0.6 \pm 0.4$ | $0.4 \pm 0.4$ | $0.9 \pm 0.3$ | -22 |
| GISS-E2-1-G | 1.3 | 1.6 | -22 | $-0.1 \pm 0.2$ | $-0.4 \pm 0.3$ | $-0.3 \pm 0.2$ | 4 |
| HadGEM3-GC31-LL | 1.0 | 1.0 | -34 | $0.1 \pm 0.3$ | $-0.1 \pm 0.3$ | $0.4 \pm 0.3$ | 4 |
| IPSL-CM6A-LR | 0.8 | 1.3 | 6 | $0.0 \pm 0.4$ | $-0.2 \pm 0.6$ | $0.6 \pm 0.2$ | -4 |
| MIROC-ES2L | 3.3 | 3.8 | -85 | $0.0 \pm 0.5$ | $-0.2 \pm 0.5$ | $0.2 \pm 0.2$ | 2 |
| NESM3 | 1.1 | 1.2 | -21 | $0.3 \pm 0.5$ | $0.0 \pm 0.5$ | $1.1 \pm 0.4$ | -7 |
| NorESM2-LM | 1.7 | 1.8 | -40 | $0.1 \pm 0.4$ | $-0.1 \pm 0.6$ | $0.4 \pm 0.1$ | -5 |
| PMIP3 model ensemble | - | - | - | $0.0 \pm 0.1$ | $-0.1 \pm 0.2$ | $0.1 \pm 0.1$ | - |
| PMIP4 model ensemble | 1.1 | 1.4 | -17 | $0.1 \pm 0.3$ | $-0.1 \pm 0.4$ | $0.5 \pm 0.1$ | -7 |
| HadCM3_127k | 1.6 | 1.6 | -3 | $0.2 \pm 0.2$ | $-0.1 \pm 0.3$ | $0.4 \pm 0.3$ | -4 |
| HadCM3_128k_H11 | - | - | - | $1.3 \pm 0.6$ | $1.1 \pm 0.7$ | $2.6 \pm 0.4$ | -40 |

$^a$ Differences as a percentage of HadISST1 levels.

$^b$ Mean differences over the Antarctic ice sheet.

$^c$ Differences as a percentage of *piControl* levels.

### 3.2 Climate anomalies at 127 ka simulated by individual models

For annual SST, mean differences between *lig127k* and *piControl* over the ocean south of 40° S range from -0.2°C (EC-Earth3-LR, Fig. 3e) to 0.6°C (FGOALS-g3, Fig. 3f; Table 3). Eight out of 12 models exhibit warmer conditions at 127 ka than preindustrial over this region (Table 3). Two models suggest colder conditions by 0.2°C (EC-Earth3-LR, Fig. 3e) and 0.1°C

(GISS-E2-1-G, Fig. 3g), respectively. While the magnitude of simulated climate anomalies at 127 ka is small, the differences





between *lig127k* and *piControl* are generally significant (Fig. 3). More than two-thirds of the models experience significant warm anomalies over the southern Indian Ocean.

For summer SST, the model responses vary between -0.7°C (EC-Earth3-LR, Fig. 4e) and 0.4°C (FGOALS-g3, Fig. 4f; Table 3). Eight models suggest colder conditions at 127 ka than preindustrial, whereas two models indicate warmer anomalies,
particularly over the southern Indian Ocean (*i.e.* ACCESS-ESM1-5, 0.2°C, Fig. 4a; and FGOALS-g3, 0.4°C, Fig. 4f).

For annual SAT over Antarctica, the model responses range from -0.3°C (GISS-E2-1-G, Fig. 5g) to 1.6°C (ACCESS-ESM1-5, Fig. 5a; Table 3). Similar to annual SST, the same ten models demonstrate warmer conditions at 127 ka relative to preindustrial, whereas the other two models, EC-Earth3-LR (-0.1°C, Fig. 5e) and GISS-E2-1-G (-0.3°C, Fig. 5g), exhibit colder conditions. Large positive annual SAT anomalies over Antarctica in ACCESS-ESM1-5 were attributed to Antarctic sea
ice reductions (King-Hei Yeung et al., 2021). However, considering the large bias (-30%) in simulating preindustrial September SIA, ACCESS-ESM1-5 might respond to 127 ka boundary conditions in the right direction but for the wrong reason.

The changes in September SIA vary between -34% (ACCESS-ESM1-5, Fig. 6a) and 4% (GISS-E2-1-G and HadGEM3-GC31-LL, Fig. 6g and 6h; Table 3). Nine models indicate reductions in September SIA (Table 3). Another model that shows expanded September SIA is MIROC-ES2L (2%, Fig. 6j), which might be explained by the already small sea ice extent in
preindustrial conditions. A few models suggest reduced September SIC over the southern Indian Ocean, such as ACCESS-ESM1-5 (Fig. 6a) and FGOALS-g3 (Fig. 6f).





**Figure 3.** Differences in annual SST between *lig127k* and *piControl* simulations. Paleo proxies in each subfigure are from Fig. 1a. We only show differences that are significant at 5% level based on the student's t-test with Benjamini-Hochberg Procedure controlling false discovery rates (Benjamini and Hochberg, 1995).





**Figure 4.** Differences in summer SST between *lig127k* and *piControl* simulations. Paleo proxies in each subfigure are from Fig. 1b. We only show differences that are significant at 5% level based on the student's t-test with Benjamini-Hochberg Procedure controlling false discovery rates (Benjamini and Hochberg, 1995).



**Figure 5.** Differences in annual SAT between *lig127k* and *piControl* simulations. Paleo proxies in each subfigure are from Fig. 1a. We only show differences that are significant at 5% level based on the student's t-test with Benjamini-Hochberg Procedure controlling false discovery rates (Benjamini and Hochberg, 1995).





**Figure 6.** Differences in September SIC between *lig127k* and *piControl* simulations. Dashed and solid black lines represent 15% contours of SIC in the *lig127k* and *piControl* simulations, respectively. Paleo proxies in each subfigure are from Fig. 1c. We only show differences that are significant at 5% level based on the student's t-test with Benjamini-Hochberg Procedure controlling false discovery rates (Benjamini and Hochberg, 1995).





### 3.3 Climate anomalies at 127 ka in reconstructions vs. individual simulations

To benchmark model performance, we introduce a Null Scenario, where the climate at 127 ka is assumed to be the same as the preindustrial climate. The concept is similar to that of a persistence forecast obtained by persisting the initial conditions in
weather forecasting (Murphy, 1992). To demonstrate a better performance than the Null Scenario, the model simulations must have a smaller RMSE when evaluated against the climate syntheses than the Null Scenario.

Reconstructed annual SST anomalies at 127 ka relative to preindustrial are underestimated by the model simulations. The reconstructed average anomalies reach 2.7±1.0°C in the Hoffman et al. (2017) dataset and 2.2±1.5°C in the Chandler and Langebroek (2021a) dataset (Table 2), whereas the simulated regional average anomalies range from -0.2±0.3°C to 0.6±0.4°C
(Table 3). Compared to the Capron et al. (2017) dataset, two-thirds of the models outperform the Null Scenario, whereas only one-third of the models achieve slightly superior performance than the Null Scenario when evaluated against Hoffman et al. (2017) and Chandler and Langebroek (2021a) datasets (Table 4). Note that the Capron et al. (2017) dataset contains only two annual SST records. The best performance is achieved by NESM3, with a minimum but still large RMSE relative to all syntheses.

For summer SST, while average anomalies in the syntheses range from 1.2±1.1°C to 2.2±1.9°C (Table 2), those in simulations vary between -0.7±0.4°C and 0.4±0.4°C (Table 3). Compared to any synthesis, only two to three models show better performance than the Null Scenario (ACCESS-ESM1-5, CNRM-CM6-1, FGOALS-g3, or NESM3, Table 4). Two-thirds of the models suggest negative summer SST anomalies, in contrast to positive anomalies revealed by all syntheses. Again, NESM3 exhibits lower RMSE by ∼0.2°C than the Null Scenario when compared to three syntheses.

For annual SAT anomalies, while the magnitude of warmth in reconstructions cannot be captured by the models (Table 2 vs. Table 3), 11 out of 12 models outperform the Null Scenario (Table 4). While the four ice core records in Capron et al. (2017) suggest warmer conditions at 127 ka than preindustrial by 2.2±1.4°C, simulated anomalies range from -0.3±0.2°C to 1.6±0.4°C. Only one model GISS-E2-1-G has a larger RMSE than the Null Scenario (2.7 vs. 2.4°C), and the best agreement is achieved by ACCESS-ESM1-5 and NESM3 with an RMSE of 1.1°C.

For September SIC, there are also large discrepancies between simulated and reconstructed anomalies. Compared with the Chadwick et al. (2021) dataset, the RMSE ranges from 40% to 56% for individual simulations, whereas the Null Scenario has an RMSE of 47%. Only one-third of models outperform the Null Scenario.



**Table 4.** RMSE between simulated and reconstructed climate anomalies at 127 ka relative to preindustrial conditions. The calculation of RMSE is introduced in Section 2.5, and the concept of the Null Scenario is explained in Section 3.3. Briefly, in the Null Scenario, we assume that the climate at 127 ka is the same as the preindustrial climate. Here EC2017 refers to Capron et al. (2017), JH2017 Hoffman et al. (2017), DC2021 Chandler and Langebroek (2021a), and MC2021 Chadwick et al. (2021).

| Models | Annual SST [°C] | | | Summer SST [°C] | | | | Annual SAT [°C] | Sep SIC [%] |
|---|---|---|---|---|---|---|---|---|---|
| | EC2017 | JH2017 | DC2021 | EC2017 | JH2017 | DC2021 | MC2021 | EC2017 | MC2021 |
| ACCESS-ESM1-5 | 4.3 | 5.7 | 2.7 | 2.7 | 3.5 | 3.1 | 1.2 | 1.1 | 41 |
| AWI-ESM-1-1-LR | 4.0 | 5.3 | 2.4 | 2.5 | 3.4 | 3.0 | 1.9 | 1.8 | 44 |
| CESM2 | 4.3 | 5.6 | 2.7 | 2.7 | 3.5 | 3.2 | 2.0 | 2.2 | 53 |
| CNRM-CM6-1 | 3.8 | 5.3 | 2.3 | 2.2 | 3.0 | 2.7 | 2.2 | 1.6 | 49 |
| EC-Earth3-LR | 4.1 | 5.5 | 2.7 | 2.8 | 3.4 | 3.4 | 2.4 | 2.3 | 48 |
| FGOALS-g3 | 3.8 | 5.4 | 2.2 | 2.3 | 3.2 | 2.7 | 1.4 | 1.8 | 40 |
| GISS-E2-1-G | 4.1 | 5.5 | 2.7 | 2.8 | 3.6 | 3.3 | 1.9 | 2.7 | 49 |
| HadGEM3-GC31-LL | 4.0 | 5.5 | 2.7 | 2.6 | 3.4 | 3.2 | 1.6 | 1.8 | 51 |
| IPSL-CM6A-LR | 4.2 | 5.6 | 2.7 | 2.6 | 3.4 | 3.1 | 2.0 | 1.7 | 44 |
| MIROC-ES2L | 4.1 | 5.6 | 2.8 | 2.8 | 3.6 | 3.1 | 1.8 | 2.1 | 47 |
| NESM3 | 3.3 | 5.2 | 2.1 | 2.1 | 3.1 | 2.7 | 1.7 | 1.1 | 50 |
| NorESM2-LM | 4.4 | 5.5 | 2.7 | 2.6 | 3.3 | 3.2 | 2.1 | 2.1 | 56 |
| PMIP3 model ensemble | 4.3 | 5.5 | 2.7 | 2.5 | 3.4 | 3.0 | 1.7 | 2.3 | - |
| PMIP4 model ensemble | 4.0 | 5.5 | 2.5 | 2.5 | 3.4 | 3.0 | 1.8 | 1.8 | 47 |
| HadCM3_127k | 3.1 | 5.2 | 2.5 | 2.4 | 3.4 | 3 | 1.9 | 2 | 43 |
| HadCM3_128k_H11 | 1.2 | 5 | 2 | 2 | 3.2 | 2.4 | 1.3 | 1.4 | 38 |
| Null Scenario | 4.2 | 5.5 | 2.6 | 2.4 | 3.2 | 2.9 | 1.6 | 2.4 | 47 |

### 3.4 Climate anomalies at 127 ka in reconstructions vs. PMIP3 and PMIP4 model ensembles

For annual SST, simulated anomalies by both model ensembles at 127 ka relative to preindustrial have a narrow range across all
core sites (-0.5 to 0.5°C), compared to those in the syntheses (-6.8 to 11.5°C, Fig. 8a and 8d). From the PMIP3 to PMIP4 model ensembles, there are increases in both the magnitude and spatial variability of annual SST anomalies, even only from 0.0±0.1°C to 0.1±0.3°C (Table 3, Fig. 7a and 7d). When evaluated against the three syntheses, the PMIP4 model ensemble outperforms the Null Scenario, whereas the PMIP3 model ensemble does not (Table 4). Due to differences in boundary conditions and the number and types of participating models, the slightly reduced model-data disagreement is insufficient to indicate an
improvement from the PMIP3 to PMIP4 model ensembles.

For summer SST anomalies at 127 ka relative to preindustrial, again, both model ensembles exhibit a narrower range (-0.9 to 0.8°C) than the data syntheses (-4.0 to 6.8°C; Fig. 8b and 8e). In contrast to reconstructed large positive regional average





summer SST anomalies at 127 ka relative to preindustrial (1.2 to 2.2°C, Table 2), both model ensembles demonstrate negative regional average anomalies (-0.1°C, Table 3). When evaluated against the data syntheses, both model ensembles show larger

RMSE than the Null Scenario (Table 4). Though, spatial patterns of the summer SST anomalies indicate larger variability in the PMIP4 than PMIP3 model ensembles (Fig. 7b and 7e; the standard deviation of simulated summer SST anomalies over the ocean south of 40° S: 0.4 vs. 0.2°C). Furthermore, the PMIP4 model ensemble demonstrates warmer conditions at 127 ka than preindustrial over the southern Indian Ocean in both annual and summer SST (Fig. 7d and e).

Regarding annual SAT, reconstructed anomalies at 127 ka relative to preindustrial range from 0.9°C at EDML to 3.3°C at

Dome F, whereas corresponding values in the PMIP4 model ensemble range from 0.6°C at EDC to 0.8°C at Dome F (Fig. 8f). From the PMIP3 to PMIP4 model ensembles, the simulated anomalies over Antarctica increase from 0.1±0.1°C to 0.5±0.1°C (Table 3), and thus the RMSE relative to the Capron et al. (2017) dataset decreases from 2.3°C to 1.8°C (Table 4). Although the warmth at 127 ka suggested by the ice core records is still underestimated by the model ensembles (Fig. 7c and 7f, Fig. 8c and 8f), both model ensembles perform better than the Null Scenario (Table 4).

When looking at sea ice, we observe a reduction in September SIA by 7% at 127 ka relative to preindustrial in the PMIP4 model ensemble (Table 3). Compared to the Chadwick et al. (2021) dataset, the PMIP4 model ensemble has the same performance as the Null Scenario (RMSE: 47%; Table 4, Fig. 8g). While the PMIP4 model ensemble suggests reductions in September SIC over the southern Indian Ocean (Fig. 7g), a lack of proxies in this region prohibits further evaluation.





**Figure 7.** Simulated and reconstructed climate anomalies at 127 ka relative to preindustrial. Different columns display (a, d, h, l) Annual SST, (b, e, i, m) Summer SST, (c, f, j, n) Annual SAT, and (g, k, o) September SIC, respectively. Different rows display (a, b, c) the PMIP3 model ensemble, (d, e, f, g) the PMIP4 model ensemble, (h, i, j, k) the HadCM3, and (l, m, n, o) the HadCM3 simulation with 0.25 $Sv$ freshwater forcing, respectively. For the PMIP4 model ensemble and the HadCM3 simulations, we only show differences that are significant at 5% level based on the student's t-test with Benjamini-Hochberg Procedure controlling false discovery rates (Benjamini and Hochberg, 1995).





**Figure 8.** Simulated and reconstructed climate anomalies at 127 ka relative to preindustrial at each core site. Different columns display (a, d, h, l) Annual SST, (b, e, i, m) Summer SST, (c, f, j, n) Annual SAT, and (g, k, o) September SIC, respectively. Different rows display (a, b, c) the PMIP3 model ensemble, (d, e, f, g) the PMIP4 model ensemble, (h, i, j, k) the HadCM3, and (l, m, n, o) the HadCM3 *lig127k* simulation with 0.25 $Sv$ freshwater forcing, respectively. Three dashed black lines are added in each subfigure to help visual interpretation: $x = 0$, $y = 0$, and $y = x$.





### 3.5 Impacts of northern ice sheet meltwater on the southern mid-to-high latitude climate

The extended 3000-year hosed H11-type LIG simulation, HadCM3_128k_H11, permits investigating the impacts of meltwater release from northern ice sheets into the North Atlantic on the southern mid-to-high latitude climate. The preindustrial simulation of HadCM3 indicates good performance when evaluated against the HadISST1 dataset (1870-1899). For both annual and summer SST, although HadCM3 shows larger RMSE with the HadISST1 dataset than the PMIP4 model ensemble, it has better performance than four state-of-the-art models (*i.e.* ACCESS-ESM1-5, EC-Earth3-LR, MIROC-ES2L, and NorESM2-LM,

Table 3). For September SIA, HadCM3 shows a smaller bias to the HadISST1 dataset (-3%) than the PMIP4 model ensemble (-17%) and 10 out of the 12 individual models (up to -85%). Thus the scientific choice to use HadCM3, whilst dictated by computation constraints, particularly the slow running speed of HadGEM3, is reasonable.

Similar to the PMIP4 model ensemble, the HadCM3_127k run shows weak responses to orbital parameters and greenhouse gas concentrations. South of 40° S, HadCM3_127k simulates positive anomalies in annual SST (0.2±0.2°C) and SAT

(0.4±0.3°C) at 127 ka compared to preindustrial (Table 3, Fig. 7h and j). As in the PMIP3 and PMIP4 model ensembles, there are negative summer SST anomalies compared to preindustrial south of 40° S in HadCM3_127k (-0.1±0.3°C), and the southern Indian Ocean shows warmer conditions (Fig. 7i). Changes in September SIA are smaller in HadCM3_127k (-4%) than the PMIP4 model ensemble (-7%, Fig. 7k).

As shown in the 1600-year version of the HadCM3_128k_H11 experiment (Holloway et al., 2018; Chadwick et al., 2023),

accounting for a 0.25 $Sv$ freshwater forcing over the North Atlantic, the HadCM3 simulation gains heat in southern polar and subpolar regions compared to preindustrial. This continues in the 3000-year version of this simulation. The ocean south of 40° S becomes warmer in HadCM3_128k_H11 than preindustrial by 1.3±0.6°C in annual SST and 1.1±0.7°C in summer SST (Table 3, Fig. 7l and m). While annual SAT over the Antarctic ice sheet is also higher by 2.6±0.4°C in HadCM3_128k_H11 than preindustrial (Table 3, Fig. 7n), September SIA is reduced by 40% (Table 3, Fig. 7o).

When evaluated against the four syntheses, HadCM3_128k_H11 has the best performance among the PMIP3 and PMIP4 model ensemble, HadCM3_127k, and the Null Scenario. Compared to the four archives, HadCM3_127k shows smaller RMSE than the Null Scenario in annual SST, annual SAT, and Sep SIC, whereas summer SST is simulated with larger deviations from the archives than the Null Scenario (Table 4). In contrast, for every proxy in the archives, HadCM3_128k_H11 shows lower RMSE than the Null Scenario (Table 4).

## 4 Discussion

### 4.1 Limitations of the LIG data syntheses

While it would be helpful to provide a single unified LIG Southern Ocean and Antarctic data synthesis for benchmarking PMIP simulations, this would be a substantial additional piece of work. In the meantime, there are minor shortcomings in the four syntheses which could be addressed in the future. Firstly, the preindustrial values were derived from a gridded dataset rather

than reconstructed from core top measurements. It would be helpful to check the implications of this approach. Secondly,





the spatial coverage of the proxies is still rather uneven. For example, a lack of southern Pacific Ocean sites will cause large uncertainties if the LIG climate were to be explored using data assimilation techniques, as was done for the Last Glacial Maximum (Tierney et al., 2020). Thirdly, some discrepancies exist in the reconstructions from the same cores between different datasets, particularly between the Hoffman et al. (2017) dataset and two other datasets in Table A1. Indeed this would be one

of the main challenges to compile a single unified synthesis. Fourth, Capron et al. (2014) and Hoffman et al. (2017) did not use consistent calibration functions for each proxy as Chandler and Langebroek (2021a) did. Lastly, Chandler and Langebroek (2021a) and Chadwick et al. (2021) provided age uncertainties and reconstruction uncertainties separately, but did not estimate uncertainties accounting for both dating and reconstruction errors. It would also be useful to revisit the calibration of Antarctic ice core temperatures in light of recent work (Sime et al., 2009). Overall, it would be most helpful if a future synthesis could

address these issues.

### 4.2 Impacts of the penultimate deglaciation on the early LIG climate

The early LIG has been suggested to be a transient climate period, particularly in response to impacts of preceding and contemporary ice sheet meltwater in the North Atlantic (Barker et al., 2019; Govin et al., 2012). Stone et al. (2016) indicated that a bipolar temperature response at 130 ka might be attributable to meltwater from northern ice sheets using a 200-year hosed

HadCM3 simulation. Govin et al. (2012) also found that freshwater forcing over the North Atlantic might explain the delay in peak interglacial conditions at 126 ka at northern high latitudes. Our 3000-year HadCM3_128k_H11 simulation indicates that equilibrated responses of the Southern Ocean and Antarctica to the North Atlantic hosing characterise better consistency with proxy records at 127 ka, in agreement with results of Holloway et al. (2018) and Chadwick et al. (2023). This work adds strong evidence to support the underlying assumption of the PMIP4 Tier 2 *lig127k-H11* simulation, *i.e.* freshwater flux associated

with H11 (∼135-128 ka, Marino et al., 2015) might be crucial for the evolution of the early LIG climate. In our simulation, while AMOC has reached a steady state after about 250 model years, the Southern Ocean surface climate does not appear to come into equilibrium until around 2500 model years (Fig. A1). This poses problems when considering how best to evaluate the CMIP6/7 models in the future against LIG data, since long-term (perhaps transient) simulations, particularly using coupled ice-sheet-climate models, may be computationally infeasible for many groups (Guarino et al., 2023).

## 5 Conclusions

The LIG climate in the high latitudes is often regarded as an analogue of the future warmer climate in these regions, even though different warming mechanisms are at play. While multiple paleo proxies suggest warmer conditions at 127 ka than preindustrial over southern mid-to-high latitudes, it has been a challenge for climate models to reproduce the warm magnitude when only accounting for greenhouse gas concentration and orbital forcings. Here we use four paleoclimate syntheses and a

range of model simulations to investigate the southern polar and subpolar warming at 127 ka.

Despite reconstruction uncertainties and regional variability, 99 reconstructions of four climate variables indicate warmer conditions at 127 ka than preindustrial over the Southern Ocean and Antarctica. For instance, regionally-averaged summer





SST anomalies at 127 ka relative to preindustrial range from 1.2±1.1°C in the more southerly sited Chadwick et al. (2021) synthesis to 2.2±1.9°C in the more northerly located Chandler and Langebroek (2021a) synthesis.

Changing from preindustrial to 127 ka boundary conditions, the model responses are generally small. There is a diversity in the magnitude and spatial patterns of simulated climate anomalies at 127 ka relative to preindustrial, and a few models exhibit unanticipated average warming or cooling tendencies contrary to what is expected from solar insolation anomalies.

    The reconstructed magnitude of warmth at 127 ka is not reproduced by the Tier 1 PMIP4 *lig127k* model simulations. These simulations are forced by orbital and greenhouse gas changes and do not take account of meltwater from the ice sheets. For 340 individual models, NESM3 has the lowest RMSE among PMIP4 models when compared to most of the reconstructed variables, whereas the RMSE are still large (*e.g.* 1.7-3.1°C for summer SST) and are comparable to those from the Null Scenario (1.6-3.2°C for summer SST). For model ensembles, although there is a small improvement from the PMIP3 to PMIP4 model ensemble (*e.g.* from 2.3 to 1.8°C in RMSE for annual SAT), their performance is still close to the Null Scenario.

    Earlier work pointed out that meltwater from northern ice sheets is crucial in the simulation of the early LIG climate (Stone 345 et al., 2016; Holloway et al., 2018; Chadwick et al., 2023). We present a 3000-year HadCM3_128k_H11 simulation to explore the role of meltwater release into the North Atlantic. While HadCM3_127k exhibits similarly weak responses to 127-ka boundary conditions as the PMIP4 simulations, HadCM3_128k_H11 produces notable southern polar and subpolar warming relative to preindustrial: south of 40° S, annual SST rises by 1.3±0.6°C, summer SST increases by 1.1±0.7°C, and September SIA reduces by 40%; over the Antarctic ice sheet, annual SAT increases by 2.6±0.4°C. The 3000-year 0.25 $Sv$ freshwater forcing 350 over the North Atlantic thus, as postulated, improves model-data agreement, as indicated by smaller RMSE with the four syntheses than other simulations. Future work could explore transient meltwater impacts on the evolution of the LIG climate in both northern and southern polar regions using a hierarchy of models.

*Code and data availability.* The Capron et al. (2017) and Hoffman et al. (2017) datasets are available from here: https://doi.org/10.1016/j. quascirev.2017.04.019. The Chandler and Langebroek (2021a) dataset is available from here: https://doi.org/10.1594/PANGAEA.938620. 355 The Chadwick et al. (2021) dataset is available from here: https://doi.org/10.1594/PANGAEA.936573. The four paleoclimate syntheses processed through the authors and the site locations are available from here: https://zenodo.org/records/11079974. The HadISST1 dataset is available from here: https://www.metoffice.gov.uk/hadobs/hadisst/. The PMIP4 model simulations are available from here: https://esgf-node. llnl.gov. The PMIP3 model ensemble is available from here: https://doi.org/10.5194/cp-9-699-2013. The HadCM3 simulation output and data analysis scripts are available from the authors upon request.

**Appendix A: Additional figure and table**



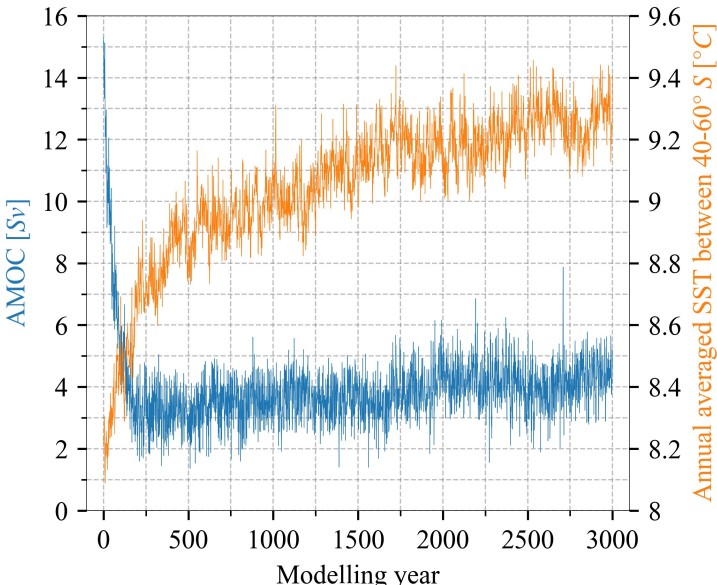

**Figure A1.** Annual values of AMOC and area-weighted mean SST between $40° S$ and $60° S$ in the 3000-year 128-ka HadCM3 simulation with 0.25 $Sv$ freshwater forcing. The AMOC is estimated as the maximum zonally and vertically integrated overturning stream function over depth at $40° N$.

*Author contributions.* EC, LCS and QG led the development of this study. RS downloaded the PMIP4 model simulations and ran the HadCM3 simulation. QG collected the data syntheses and the PMIP3 model ensemble. QG performed all data analysis and wrote with EC and RR the first draft of this manuscript. All authors contributed to the final draft.

*Competing interests.* The authors declare that they have no conflict of interest.

*Acknowledgements.* This publication was generated in the frame of DEEPICE project. The project has received funding from the European Union's Horizon 2020 research and innovation programme under the Marie Sklodowska-Curie grant agreement No 955750. EC acknowledges the financial support from the French National Research Agency under the "Programme d'Investissements d'Avenir" (ANR-19-MPGA-0001), through the Make Our Planet Great Again HOTCLIM project. We also acknowledge the PMIP4 modeling groups that contributed *lig127k* simulations.



**Table A1.** Marine sediment cores used in more than one dataset. Reconstructed anomalies at 127 ka relative to preindustrial are provided together with two-sigma errors (if available). Reconstruction methods are given in parentheses, where A refers to alkenones, F foraminifera assemblage transfer functions, N the percentage of Neogloboquadrina pachyderma sinistral, D diatom assemblage transfer functions, and M foraminiferal Mg/Ca ratios.

|  | Cores | Latitude | Longitude | Capron et al. (2017) | Hoffman et al. (2017) | Chandler and Langebroek (2021a) |
|---|---|---|---|---|---|---|
| Annual SST | MD97-2121 | -40.00 | 177.00 | 5.1 ± 3.6 (A) | 7.0 ± 1.7 (A) | 2.5 (A) |
|  | DSDP-594 | -45.50 | 174.95 | 2.9 ± 2.4 (F) | 0.5 ± 3.4 (F) | - |
|  | MD88-770 | -46.02 | 96.45 | - | 1.9 ± 2.1 (N and D) | 2.2 (F) |
|  | MD97-2120 | -45.53 | 174.93 | - | 7.7 ± 2.1 (M) | 2.8 (M) |
| Summer SST | MD94-101 | -42.50 | 79.42 | -0.2 ± 4.3 (F) | 2.8 ± 2.5 (N) | - |
|  | PS-2489-2 | -42.52 | 8.58 | -0.1 ± 2.9 (F) | 1.0 ± 3.2 (F) | 0.7 (F) |
|  | MD94-102 | -43.50 | 79.83 | 0.9 ± 4.2 (F) | 5.4 ± 2.5 (N) | - |
|  | MD88-770 | -46.02 | 96.45 | 1.6 ± 2.9 (F) | 0.5 ± 1.9 (N and D) | - |
|  | MD02-2488 | -46.49 | 88.02 | 4.0 ± 3.9 (F) | 3.8 ± 2.9 (N) | 2.6 (F) |
|  | MD84-551 | -55.01 | 73.28 | 3.4 ± 2.3 (F) | - | 3.9 (D) |
|  | MD97-2120 | -45.53 | 174.93 | 1.1 ± 1.7 (M) | - | 2.7 (A) |
|  | MD97-2121 | -40.00 | 177.00 | 3.4 ± 2.6 (A) | - | 2.2 (A) |
|  | PS-2102-2 | -53.07 | -4.99 | 0.9 ± 0.8 (D) | - | 0.8 (D) |
|  | SO136-111 | -56.40 | 160.14 | -0.1 ± 3.8 (D) | - | 0 (D) |

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
