# Peer review of "Assessment of the southern polar and subpolar warming in the PMIP4 Last Interglacial simulations using paleoclimate data syntheses"

_EGUsphere, 2024_

## Author Comment (AC1)

**Responses to comments of Referee #1**

The study by Gao et al. performed a 3000-yr 128-ka simulation to highlight the importance of the meltwater release into the North Atlantic in reducing the model–data discrepancies over the Southern Ocean. The manuscript is clearly written, and the topic fits the CP. However, there are several concerns that should be addressed.

Response: We appreciate the time of Referee #1 in reviewing this manuscript, thank you. These are useful discussions. Please check our detailed responses below.

Major comments:

1. In section 2.2, the authors declared that they selected the most recent surface temperature data syntheses and SIC dataset. I am wondering the reason for ignoring the data from Turney et al. (2020), as they also provide annual SST data during the early LIG. If there is no special reason, comparisons between the SST data from Turney et al. (2020) and models should be added.

[Reference]

Turney, C. S. M., et al. (2020). A global mean sea-surface temperature dataset for the Last Interglacial (129–116 kyr) and contribution of thermal expansion to sea-level change. Earth System Science Data 12(4): 3341-3356.

Response: We thank the reviewer 1 for their suggestion and we will explain below why we think that we would favour not presenting a model-data comparison exercise at 127 ka that include the Turney et al. (2020) dataset

The major issue of the Turney et al. (2020) is related to the fact that the authors present a data compilation based on paleoclimatic records kept on the original chronologies. They do not attempt to build a coherent temporal framework between the different paleoclimatic records nor to provide an assessment of the large chronology errors that are associated with marine sediment records across the LIG. This is hugely problematic considering that using different dating strategies for paleorecords across the LIG could lead to age discrepancies of up to 6 ka, as detailed in Govin et al. (2015). Also, it goes against the large efforts put together over the past years to guide the community towards being careful with harmonizing age models for paleoclimatic archives during the Last Interglacial. Indeed, it is widely recognized now how key it is to harmonize paleorecord chronologies when building data compilations in order to provide (1) a realistic representation of the LIG climate and (2) appropriate benchmarks to evaluate the LIG model simulations (e.g. Capron et al. 2014, Govin et al. 2015, Stone et al. 2016, Hoffman et al. 2017, Capron et al. 2017, Otto-Bliesner et al. 2017, Menviel et al. 2019).

In other words, we would like to strengthen the fact for the purpose of our work there is no added value of the Turney et al. (2020) dataset relative to the existing global SST compilation proposed by Hoffman et al. (2017) based on consistent timescales. In addition, the Turney et al. (2020) synthesis was not intended to be a time slice. We think

showing a model-data comparison for the Turney et al. (2020) syntheses would be a step-back for the data communities working on improving the spatio-temporal representation of the LIG climate as it ignores the current understanding of chronological uncertainties for the LIG and it would lead to misinterpretation and misuse of this synthesis by other scientists part of the model community who might not necessary familiar with chronology-related subtleties, when performing model-data comparisons.

Finally, we would like to refer to Capron et al. (2017) and Otto-Bliesner et al. (2021) for further discussion on why peak-warmth climate synthesis over the LIG should be avoided for the model-data comparisons taking place in the framework of the PMIP4 lig simulations. At the time, we discussed the Turney and Jones 2010 and McKay et al. 2010 dataset and our points are still valid and apply to the peak-warmth centered LIG dataset from Turney et al. (2020). We are now clearer on the limitations of the peak-warmth-centered data syntheses in the revised manuscript.

We added the following sentence to line 35:

"Subsequently, Turney et al. (2020) compiled the maximum annual SST between 129-124 ka from 189 marine sediment and coral records. However, they still used the original age models and assumed global synchronous peak warming conditions, which, as mentioned, limit its applicability for the evaluation of equilibrium model simulations at a specific date across the LIG."

For completeness, we gathered the maximum annual SST estimates during the early LIG (129-124ka) from Turney et al. (2020) as in following Fig 1. There are 28 records south of 40 degrees south. There are more records in the Turney dataset compared to the ones we used as less strict criteria were applied. The anomalies are relative to HadISST 1981-2010 and the average anomalies are 2.5°C. The RMSE between LIG climate anomalies in the Turney dataset and the Null Scenario, HadCM3_127k, and HadCM3_128k_H11 are 3.8°C, 3.6°C, and 3.0°C, respectively (though the simulated anomalies are relative to preindustrial values). Since HadCM3_127k performs similarly as the Null Scenario and HadCM3_128k_H11 has the smallest RMSE, it complies with the results obtained when performing the model-data comparison with the other data syntheses. However, as mentioned above, the use of the Turney et al. (2020) synthesis is associated with large problems with chronology and prevents us from drawing any robust conclusions.

[Figure]

**Figure 1**: Reconstructed maximum annual SST anomalies during 129-124 ka relative to 1981-2010 over the Southern Ocean from Turney et al. (2020).

2. In section 2.3, the experimental setups are not clear. The boundary conditions and external forcings (e.g. greenhouse gases, orbital parameters, ice sheet, vegetation, land-sea mask) at 128 ka need to be displayed or indicated directly.

Response: We added in the revised manuscript the following description to section 2.3, Line 159:

The greenhouse gas concentrations in this simulation are close to those set by the PMIP4 lig127k guideline: carbon dioxide at 275 parts per million (ppm), methane at 706.8 parts per billion (ppb), and nitrous oxide at 266 ppb. The vegetation, aerosol, and ice sheets were set identical to the corresponding preindustrial simulation (Tindall et al., 2009).

3. In section 3.1, the authors evaluated the performance of PMIP4 models in simulating SST over the Southern Ocean, but they only provide the RMSE. The spatial correlation coefficients between PMIP4 models and HadISST1 dataset should also be provided. I suggest providing a Taylor diagram to show the performance of PMIP4 models more clearly.

Response: Thank you for the suggestion on a Taylor diagram. It does provide more information on correlation coefficients and standard deviations as shown in the following figure 2 for annual mean SST south of 40 degree south, but we chose to display area-weighted RMSE as it is for three reasons:

1) Compared to the RMSE in the Taylor diagram, we consider our area-weighted RMSE to be more robust, as it takes into account meridional variations in the area of grid cells.

2) As meridional gradients in annual mean SST are well captured by all models, all correlation coefficients are larger than 0.93. This means the Taylor diagram provides little insight into any model bias.

3) Except for MIROC-ES2L, all other models demonstrate similar RMSE and standard deviation. Since this can be expected from the  large warm bias shown in Fig 2j, additional presentation of this similar information may be redundant.

However, if the referee and/or the editor still would like us to include it in the revised manuscript, we can do so.

[Figure]

**Figure 2**: A Taylor diagram showing standard deviation, root mean squared errors, and correlation coefficients between annual mean SST south of 40 degree south from 12 models and the HadISST1 dataset.

4. Across the manuscript, the authors suggested that the orbital parameters, greenhouse gases and Antarctic ice sheet played a limited role in the warming of the Southern Ocean during the early LIG, and attributed the warming to the meltwater release into the North Atlantic. Zhang et al. (2023) indicated that the global sea level rising during the LIG (at 126 ka) can also reduce the model-data discrepancies over the Southern Ocean. I suggest that more discussions about this point need to be added.

[Reference]

Zhang, Z., et al. (2023). Atmospheric and oceanic circulation altered by global mean sea-level rise. Nature Geoscience 16(4): 321-327.

Response: We are grateful for being pointed to this nice article. The following sentence is added into the Introduction at Line 51.

"While Zhang et al. (2023) found that increased global mean sea level warms southern mid-to-high latitudes at 126 ka using a climate model NorESM1-F, the root mean squared errors (RMSE) between temperature anomalies at 126 ka relative to preindustrial from the simulations and the Chandler and Langebroek (2021a) dataset were only reduced by ~10% while applying a 5-m or 10-m sea level rise."

5. The authors indicated that long-time simulation (i.e. the 3000-yr simulation) of H11 is likely required to capture the full magnitude of the Southern Ocean and Antarctic warming, following the guideline of the modeled linear SST trend in a 1600-yr simulation by Holloway et al. (2018). In Figure A1, the linear trend of the SST from 0 to 1600 years is significant. However, from 1600 to 3000 years, the SST seems to fluctuate near a mean state, rather than shows a long-term increasing trend. So I am wondering the necessity of running a long-time simulation in reconciling the model-data mismatch based on the current results. I suggest providing the difference between the "short" simulation and "long" simulation at 128 ka to highlight the novelty of long-time simulation performed in this study.

Response: We agree that the increasing rate of Southern Ocean SST slows down after ~1600 model years. This indeed contrasts the expectation of Holloway et al. (2018) who assumed a linear trend in temperature changes and proposed that 3-to-4 thousand years simulation may reconcile the model-data mismatch in ice core records. Our preliminary analysis using a 2000-year simulation indeed provides similar results as the 3000-year simulation. Though, we still consider it valuable to reach a quasi-equilibrium state, which facilitates the investigation of post-hosing abrupt climate changes. This is actually a main research topic in our group now. We added the following sentences in Line 312:

"In addition, we note that the rate of Southern Ocean SST increase slows down after ~1600 model years (Fig. A1), and our preliminary analysis using a 2000-year HadCM3_128k_H11 simulation gives similar results as the 3000-year one. This contrasts with the expectation of Holloway et al. (2018) that the Southern Ocean and Antarctica would exhibit a linear warming trend throughout the period of meltwater input. However, we consider it valuable to run our model long enough to reach a quasi-equilibrium state to test their hypothesis. It also facilitates the investigation of post-hosing abrupt climate changes."

Minor comments:

Line 87: Four most recent surface temperature data syntheses, may be three?

Response: It is four since Chadwick et al. (2021) also provided temperature reconstructions.

---

## Author Comment (AC2)

**Responses to comments of Referee #2**

Last interglacial climate simulations provide an interesting insight into a warmer climate state, albeit under different orbital and greenhouse gas forcings than those of present-day. Here the authors address a notable issue in the PMIP4 last interglacial simulations, which show very little Southern Ocean SST warming in comparison to proxy records. The limited improvement between PMIP3 and PMIP4 in that respect highlights the work still to be done by both the proxy and modelling communities to resolve these differences. However, by adding additional freshwater into the North Atlantic in a classic hosing experiment, the authors find considerable improvement in the simulation of last interglacial Southern Ocean SST, at least in their model (HadCM3).

Response: Thank you for your valuable time in reviewing this manuscript. We have considered each comment carefully and responded to each one as below.

This is an interesting experiment fitting the scope of CP, and is of particular relevance given increasing interest in hosing experiments for both future and past climates. Nevertheless there are some important weaknesses in the design of the experiment that I believe should be addressed, as follows.

(1) Antarctica will also have contributed anomalous freshwater fluxes during the LIG. These may actually act against your northern influence, for example by cooling SST (Mackie et al. 2020 and several similar studies). Recent work has suggested Antarctica's sea-level contribution was early in the LIG, e.g. before 126 ka (Barnett et al, 2023). In that case the Antarctic freshwater fluxes may have been just as important as North Atlantic freshwater fluxes in their influence of Southern Ocean SST. Even though I understand the logistical advantage of simply continuing an existing model run, and the computational constraints, this point needs some careful discussion in your manuscript as it is (in my view) a major weakness of your experiment. How do the magnitudes of Arctic freshwater-induced cooling compare with Antarctic freshwater-induced warming in the Southern Ocean? Previous modelling studies can help answer this. Bearing in mind that sea-level rise could also help explain the apparent cold bias in PMIP4 lig127k Southern Ocean SST (Zhang et al. 2023).

Response: We really appreciate these insights and we fully agree that our idealised scenario does not account for the full range of processes. We added the following paragraph in the Conclusion at Line 356 to discuss potential future work on this.

"It is important to note that the hosed 128-ka HadCM3 simulation only represents an idealised scenario. The impacts of ice sheet meltwater would depend on its location, magnitude, and timing (He and Clark, 2022), as well as on climate background states (Pöppelmeier et al., 2023; Lynch-Stieglitz et al., 2014). Given that the Antarctic ice sheet may contribute to the peak early LIG global sea level (Barnett et al., 2023), the corresponding freshwater input could also influence the early LIG climate (Mackie et al., 2020). These processes should be investigated with coupled ice-sheet-climate models."

We added a sentence in Introduction at Line 51 to discuss the study of Zhang et al. (2023).

"While Zhang et al. (2023) found that increased global mean sea level warms southern mid-to-high latitudes at 126 ka using a climate model NorESM1-F, the root mean squared errors (RMSE) between temperature anomalies at 126 ka relative to preindustrial from the simulations and the Chandler and Langebroek (2021a) dataset were only reduced by ~10% while applying a 5-m or 10-m sea level rise."

(2) Comparison of simulated and reconstructed SST. See Section 2.5. Here the implication is that the simulated SST is being tested against the "truth", which here comprises SST reconstructions. However, reconstructions themselves have considerable uncertainty as discussed in Chandler & Langebroek (2021a,b). Their recommendation was to focus comparison on regional averages, rather than site-by-site. I would suggest to follow that approach in this paper and (for example) use the proxies to get three regional SST anomalies (Atlantic, Indian, Pacific sectors) then evaluate your results on a regional basis rather than site-by-site basis.

Response: We fully agree that the reconstructions may deviate from the "ground truth". Indeed we also compared regional averages between Table 2 and 3 (LIG-PI columns). And by calculating RMSE across sites, it takes into account regional variations, though it is not regional averages.

For preliminary analysis we also calculated regional averages for southern Atlantic, Indian, and Pacific oceans for model output. However, we chose not to show them for two reasons: 1) The differences between different ocean sectors in model-data comparisons are small. 2) Each ocean sector contains only a few records. Splitting available records into three subsets may introduce statistical bias in subsequent evaluation.

(3) Use of HadISST1 1870-1899 as a PI reference dataset. Probably this has some precedent in other studies, but as acknowledged in Section 2.4 there are very sparse observations for the Southern Ocean during 1870-1899. Consequently this comparison is not very convincing or useful without a lot more information about the errors in this region, in HadISST1. I suspect they will be fairly substantial! Why not use a more recent period? For example CMIP historical simulations. I'm fairly sure the models contributing to lig127k all have a historical simulation using the same configuration as PI.

Response: We understand the concerns on the quality of HadISST1 over the Southern Ocean during the preindustrial period. As suggested, we compared CMIP6 historical simulations of the 12 models with the SST product from the European Space Agency Climate Change Initiative (ESACCI) version 2.1 (1982-2014). As shown in the following figure 3, the model bias is quite similar to that while comparing piControl simulations with HadISST1 (Fig. 2). Considering that the data syntheses estimated LIG climate anomalies relative to preindustrial, we show the comparison between piControl simulations and HadISST1 in the manuscript.

[Figure]

**Figure 3**: Differences in annual SST between historical simulations by the 12 models listed in Table 1 and the ESACCI SST dataset version 2.1 (1982-2014).

(4) Again in the comparison with reconstructions, the Turney et al 2020 ESSD paper is a surprising omission here. At first it might seem the Turney et al. methodology makes direct comparison with lig127k a bit trickier (Turney et al. report a "LIG average", not a time slice). However, they also report a 129-124 ka peak warmth. I would suggest using this, since it's actually very similar to the Capron et al. methodology in practice. This is because Capron et al. aligned their SST records to EDC air temperature, under the assumption of synchronous changes across the entire region. Consequently, their SST peaks are synchronous across all records and so the peak warmth in their study is always at 128 ka. This lies within the 129-124 ka window used by Turney et al. If you prefer to avoid too many comparisons, maybe swap the Capron et al dataset for the newer Turney et al. dataset.

Response: We thank Reviewer 3 for their suggestion and we now refer to the Turney et al. (2020) dataset in the revised version of the paper. We also explain below why we think that the Turney et al. (2020) dataset is not appropriate for our model-data comparison exercise at 127 ka (much of this argumentation is also provided to answer comments from Reviewer 1 and Reviewer 3 on this topic).

The major issue of the Turney et al. (2020) is related to the fact that the authors present a data compilation based on paleoclimatic records kept on the original chronologies. They do not attempt to build a coherent temporal framework between the different paleoclimatic records nor to provide an assessment of the large chronology errors that are associated to marine sediment records across the LIG. This is hugely problematic considering that using different dating strategies for paleorecords across the LIG could lead to age discrepancies of up to 6 ka as detailed in Govin et al. (2015). Hence, their strategy goes against the large efforts put together over the past years to guide the community towards being careful with age models for paleoclimatic archives during the LIG. Indeed, it is widely recognized now how key it is to harmonize paleorecord chronologies when building data compilations in order to provide (1) a realistic representation of the LIG climate and (2) appropriate benchmarks to evaluate the LIG model simulations (e.g. Capron et al. 2014, Govin et al. 2015, Stone et al. 2016, Hoffman et al. 2017, Capron et al. 2017, Otto-Bliesner et al. 2017, Menviel et al. 2019).

As a result showing a model-data comparison for this dataset would be a step-back for the data communities working on improving the spatio-temporal representation of the LIG climate since as previously mentioned, it ignores the current understanding of chronological uncertainties for the LIG and it would lead to misinterpretation and misused of this synthesis by other scientists part of the model community who might not necessary familiar with chronology-related subtleties, when performing model-data comparisons for the LIG.

Having said that we are aware that the syntheses that we use in our manuscript for the model-data comparisons still are attached to some limitations and improved syntheses should be developed in the future. We stress this point more clearly in the revised manuscript.

Other points follow below.

L5: are all 99 reconstructions independent, i.e., are they 99 different proxy records?

Response: No, some of the reconstructions use the same marine sediment cores as listed in Table A1. This is indicated at Line 135.

"Among these 99 records, there are 20 pairs of reconstructions based on the same marine sediment cores in at least two different datasets (Table A1)."

L10: Again from earlier comment, note that anomalous FW into SO might contribute some cooling (Mackie et al. 2020 & many others)

Response: We fully agree and it is stressed in the Conclusion.

"It is important to note that the hosed 128-ka HadCM3 simulation only represents an idealised scenario. The impacts of ice sheet meltwater would depend on its location, magnitude, and timing (He and Clark, 2022; Roche et al., 2010), as well as on climate background states (Pöppelmeier et al., 2023; Lynch-Stieglitz et al., 2014). Given that the

Antarctic ice sheet may contribute to the peak early LIG global sea level (Barnett et al., 2023), the corresponding freshwater input could also influence the early LIG climate (Mackie et al., 2020). These processes should be investigated with coupled ice-sheet-climate models."

L30, L88: I'd agree that estimating LIG warmth by compiling peak temperatures is not an appropriate approach. However, the SST records used by Capron et al. 2014/2017 were aligned to the EDC ice core temperature record, such that their synthesis also implicitly represents a synthesis of 'peak warmth' as noted in my earlier comment. Hoffman et al. followed a mixed approach (three key records representing three main ocean basins were aligned to EDC, but other records in each basin aligned to key record by d18O. Specifically from Capron et al: "*Marine records are transferred onto AICC2012 by assuming that surface-water temperature changes in the sub-Antarctic zone of the Southern Ocean (respectively in the North Atlantic) occurred simultaneously with air temperature variations above Antarctica (respectively Greenland)*".

Response: We acknowledge that making climate assumptions to infer age models is not ideal. However, in the case of the assumptions that Capron et al. (2014, 2017) used for the hemispheric alignments, it appears to be reasonable since the timing of climatic changes and the synchronicity between those observed in the ocean and in the ice core records could be checked while the different records were dated independently ((see Capron et al. 2014). Still, we propose to add the following sentence at Line 324 in Section 4.1 to stress this limitation.

"Lastly, it would be beneficial to build a synthesis with a coherent chronology independent from climate assumptions."

As mentioned before, we avoided using the Turney et al. (2020) dataset as they did not use consistent chronology and the peak warmth may occur much earlier than 127 ka. For comparison, Capron et al. (2014) found that peak temperatures occur at 129.3±0.9 ka in the Southern Hemisphere.

L118: Extraction of 127 ka anomaly. Why use 128 ka for the 127 ka anomaly? Surely better to use an average of 128 and 126 ka?

Response: We considered to use 128-ka values (20 and 21 records for annual and summer temperature, respectively), 126-ka values (15 and 17 records), and the average of 128- and 126-ka values (11 and 13 records). The average annual and summer temperature anomalies are similar: 2.2°C and 2.2°C for 128-ka, 2.4°C and 2.4°C for 126-ka, 2.5°C and 2.5°C for the average of 128- and 126-ka, respectively. We decided to use 128-ka values as a conservative estimate of the temperature anomalies with more records.

L145: Different characteristics of Chadwick et al 2021 might also reflect that they only used diatoms, whereas other datasets used multiple proxies.

Response: We added this point:

"The Chadwick et al. (2021) dataset suggests the smallest regional summer SST anomaly (1.2±1.1°C), which might result from the more southerly site locations (Fig. 1b) **and the fact that they used only diatoms for reconstructions**."

L156: I'd suggest to recap what are the key parameters/modelling choices used by Holloway et al. 2016 – presumably you keep these the same?

Response: Yes, we added the following sentence:

"The greenhouse gas concentrations in this simulation are close to those set by the PMIP4 lig127k guideline: carbon dioxide at 275 parts per million (ppm), methane at 706.8 parts per billion (ppb), and nitrous oxide at 266 ppb. The vegetation, aerosol, and ice sheets were set identical to the corresponding preindustrial simulation (Tindall et al., 2009)."

Fig 1: Can this be split into four rows, for the four studies, otherwise symbols plot over each other in a jumble.

Response: We agree that different records overlap in Fig. 1. We did try to plot each synthesis separately, but it does not look much better because records in the same synthesis also overlap.

Table 2 caption: useful to specify here again what is the PI reference used for the temperature anomalies.

Response: Added: "The preindustrial reference values are derived from HadISST1."

L164 'afflicted'... 'affected'?

Response: Changed.

L181: largest error, not largest bias. RMSE and bias are not the same thing. Which is an important point: you are ranking the datasets on their RMSE, not their bias. I think the bias should also be reported along with the RMSE.

Response: We changed "largest bias" to "largest RMSE". We focused on qualitative positive/negative bias, rather than delving into details of quantitative differences.

Table 3: what are the reported error bounds in the lig127k vs piControl anomalies?

Response: These are the standard deviations of the temperature anomalies over the region south of 40 degree south. We modified the caption:

"One standard deviation of the temperature anomalies over the region is given after mean temperature differences."

Table 4 headings: suggest to include RMSE specificically in the headings, i.e, Annual SST RMSE, Summer SST RMSE, etc otherwise this looks like a table of actual temperatures rather than temperature errors.

Response: Added.

Figs 3,4,5,6: "We only show differences that are significant at 5% level based on the student's t-test with Benjamini-Hochberg Procedure controlling false discovery rates (Benjamini and Hochberg, 1995)." What does this refer to? I can't see where differences are illustrated.

Response: "differences" is changed to "temperature/sea ice anomalies"

L218 Null scenario: ("*To benchmark model performance, we introduce a Null Scenario, where the climate at 127 ka is assumed to be the same as the preindustrial climate. ... To demonstrate a better performance than the Null Scenario, the model simulations must have a smaller RMSE when evaluated against the climate syntheses than the Null Scenario*"). Can demonstration of the 'Null Scenario' be defined more clearly? In particular there could be some confusion here about what is being compared with what. Is the Null Scenario the piControl simulation compared with HadISST1? This approach is problematic because of potentially large errors in HadISST1 1870-1899 PI as noted above for Sec 3.1. Overall this confusion makes subsequent discussion somewhat difficult to evaluate.

Response: Thank you and no, the Null scenario is not the piControl simulation compared with HadISST1. The text is modified to clarify it:

"To benchmark model performance, we introduce a Null Scenario, where the climate at 127 ka is assumed to be the same as the preindustrial climate. **Therefore, in the Null Scenario, temperature and sea ice anomalies at 127 ka relative to preindustrial are zero.** The concept is similar to that of a persistence forecast obtained by persisting the initial conditions in weather forecasting (Murphy, 1992). To demonstrate a better performance than the Null Scenario, **i.e. to be useful,** the model simulations must have a smaller RMSE when evaluated against the climate syntheses than the Null Scenario."

Fig 8: vertical error bars for the PMIP3 and PMIP4 ensembles would be useful, e.g. to show the standard deviation.

Response: This is a very good idea. The following figure 4 is an example for the PMIP4 ensemble of summer SST. The vertical blue lines show one standard deviation across 12 individual models. As the inter-model spread is relatively small, one standard deviation is normally around 0.5 degree. We added this for PMIP4 model ensembles, but not for PMIP3 model ensemble as individual model simulations are not available for PMIP3.

[Figure]

Figure 4. Reconstructed and simulated summer SST from the PMIP4 ensemble. The vertical blue lines represent one standard deviation among 12 models.

L272: Again evaluation against 1870-1899 HadISST1 is not a good benchmark given the likely errors in that dataset (particularly lack of data in S Ocean) & I would suggest to evaluate CMIP historical simulations against more recent observations in

Response: Please check the response above.

L286: "The ocean south of 40S becomes warmer...". Here and possibly several other places it is better to stick to "SST" rather than "ocean temperature" as it's only the sea-surface temperature that is being analysed. Suggest: "SST south of 40S becomes warmer..." etc.

Response: Thank you. It is changed as suggested.

L299: Pros/cons of using core-top reconstructed SST or observed SST ... Of particular relevance here, not all the LIG SST reconstructions have a core-top sample.

Response: Modified: "Firstly, the preindustrial values were derived from a gridded dataset rather than reconstructed from core top measurements, **which are not available for some cores**."

L368: Acknowledgements – as well as acknowledging PMIP contributors, maybe nice to also acknowledge the many authors who have made their SST reconstructions available on public repositories.

Response: yes, that is true. We added: We acknowledge the PMIP modeling groups that contributed LIG simulations, the authors who generated the four palaeoclimate syntheses, and all scientists who produced climate reconstructions from paleoclimatic archives.

Refs

Barnett et al. 2023, Science, https://doi.org/10.1126/sciadv.adf0198.

Capron et al. 2017, QSR, https://doi.org/10.1016/j.quascirev.2017.04.019.

Chandler & Langebroek (2021a), QSR, https://doi.org/10.1016/j.quascirev.2021.107190.

Chandler & Langebroek (2021b), QSR https://doi.org/10.1016/j.quascirev.2021.107191.

Mackie et al. 2020, J. Clim, https://doi.org/10.1175/JCLI-D-19-0881.1.

Turney et al. 2020, ESSD, https://doi.org/10.5194/essd-12-3341-2020.

Zhang et al. 2023, Nat Geos, https://doi.org/10.1038/s41561-023-01153-y.

---

## Author Comment (AC3)

**Responses to comments of Referee #3**

Gao et al compared PMIP4 simulations for the Last Interglacial (LIG, 127 ka) with existing paleoclimate syntheses of sea and air temperatures, and sea ice concentration. The authors found that the warming recorded in the paleoclimate data cannot be captured by LIG model simulations. Aiming to explain the large model-data discrepancy, the authors also performed a North Atlantic freshwater hosing simulation on 128 ka and found that this simulation better agrees with the paleoclimate data syntheses.

The authors are trying to reconcile the data-model discrepancy for the LIG, which is suitable for Climate of the Past. However, both parts of the study have some major issues that need to be addressed before being considered for publication.

Response: We are thankful for your constructive comments. Please check below the responses to each comment.

Data syntheses

The authors acknowledge some limitations of using different LIG data synthesis (Sec 4.1). However, the quality of the employed data syntheses (mainly SSTs) needs to be substantially improved to make the model-data comparison meaningful. In the supplementary data table, 127-ka annual SST anomaly appears to be unrealistic (11.5 C at site ODP 1089, >5 C at E49-17 and MD02-2588, -6.8 C at MD73-025, 7.7C at MD97-2120). The SST anomalies at the same site based on different studies are drastically different (e.g., DSDP 594, MD88-770, MD97-2120, etc).

Response: We fully understand the concerns on the quality of the employed data syntheses. We noted the large SST anomalies in the Hoffmann et al. (2017) dataset, and the large differences in the reconstructions from the same cores between this dataset and other two datasets. This indeed motivated us to adopt multiple syntheses and focus on regional averages as also recommended in Capron et al. 2017. We still consider the adopted strategy here to be effective for model-data comparison. In fact the PMIP4 community thought so as well while recommending the use of these dataset for the lig127k model-data comparison exercises (Otto-Bliesner et al. 2017). We agree with Reviewer 3 that a single synthesis resolving all the issues is highly desirable, however 1) this is beyond the scope of our study and 2) we are aware of a group of experts already working on it at the moment.

The quality of the data syntheses can be significantly improved by revisiting the original data to resolve potential issues associated with inconsistent age models, different proxy calibrations, and the core-top SST values. These issues, mentioned by the authors, needed to be addressed.

Response: We fully agree that the data syntheses we looked into in our manuscript could be further improved and we are aware of on-going initiatives led by another group investigating the different proxy calibrations and the core-top SST values to build a new LIG data synthesis. Still, significant improvements in the syntheses we are using

have been made (in particular related to the age models) while detailed PMIP4 model-data comparisons have not been made yet for individual models. Hence, we are convinced that our  model-data comparisons with this particular selection of data syntheses is of added value. As a matter of fact, the data syntheses that we are using are the ones that have been recommended by the PMIP community itself (please see the recommendations formulated by Otto-Bliesner et al. 2017). In addition, we consider that it is already a great improvement compared to what had been done as part of the PMIP3 exercise where only peak-centred data syntheses could be used to evaluate the model simulations (Lunt et al. 2013). In the revised manuscript, we try and stress more clearly the limitations of the current data synthesis and the need for a data synthesis that will tackle some of the identified limitations in Section 4.1 Limitations of the LIG data syntheses:

 While it would be helpful to provide a single unified LIG Southern Ocean and Antarctic data synthesis for benchmarking PMIP simulations, this would be a substantial additional piece of work. In the meantime, there are minor shortcomings in the four syntheses which could be addressed in the future. Firstly, the preindustrial values were derived from a gridded dataset rather than reconstructed from core top measurements, which are not available for some cores. It would be helpful to check the implications of this approach. Secondly, the spatial coverage of the proxies is still rather uneven. For example, a lack of southern Pacific Ocean sites will cause large uncertainties if the LIG climate were to be explored using data assimilation techniques, as was done for the Last Glacial Maximum (Tierney et al., 2020). Thirdly, some discrepancies exist in the reconstructions from the same cores between different datasets, particularly between the Hoffman et al. (2017) dataset and two other datasets in Table A1. Indeed this would be one of the main challenges to compile a single unified synthesis. Fourth, Capron et al. (2014) and Hoffman et al. (2017) did not use consistent calibration functions for each proxy as Chandler and Langebroek (2021a) did. Then, Chandler and Langebroek (2021a) and Chadwick et al. (2021) provided age uncertainties and reconstruction uncertainties separately, but did not estimate uncertainties accounting for both dating and reconstruction errors. It would also be useful to revisit the calibration of Antarctic ice core temperatures in light of recent work (Sime et al., 2009). Lastly, it would be beneficial to build a synthesis with a coherent chronology independent from climate assumptions. Overall, it would be most helpful if a future synthesis could address these issues.

The SST reconstruction at any point e.g., at 127 ka can be subject to uncertainties associated with measurements. Using the average over a period (e.g., 125-128 ka) can reduce the impact of such an influence.

Response: It is true that all reconstructions are associated with dating and reconstruction uncertainties. This was considered by both Capron et al. (2017) and Hoffmann et al. (2017) to provide quantified uncertainties associated to both the dating and the method used for SST reconstructions. In addition we would like to note that the reconstructions of Capron et al. (2017) at 127 ka represent an average value  between

126-128 ka. We are not sure at this stage whether a longer average period would be better, since 127-ka was experiencing millennial-scale climate changes.

To derive the 127-ka SST anomaly compared to pre-industrial from paleo data, Holocene SST changes need to be considered too. This is because the core-top ages at many sites are not late Holocene.

Response: We are sorry we are not sure to understand the comment from the reviewer. However we would like to mention that we are aware that core top SST reconstruction might diverge from the PI values inferred from HadISST and this may introduce systematic offsets in temperature anomaly calculations. However, it was noted in the studies publishing the data synthesis that using core top SST as PI reference is complicated by the perturbation or the loss of the most recent sediments during the coring procedure, making it very difficult to date the core-top.

For SSTs, I suggest focusing on 1 data synthesis taking some of the above issues into account, and adding more sites following the same criteria.

These are additional work but are necessary to make the model-data comparison meaningful.

Response: As mentioned, while we agree a future synthesis addressing all the issues is desired as discussed in section 4.1, the current model-data comparison still provides valuable insights. For example, all four syntheses suggest a warmer Southern Ocean and Antarctic at 127 ka, and all model simulations without hosing cannot reproduce the magnitude of warming. This contrast indicates a robust finding that some processes are missing in the model simulations. Using a hosed simulation, we argue that meltwater input from northern ice sheets can be a candidate to explain the model-data mismatch.

In addition, since any new LIG data synthesis will likely be associated with some subjective decisions, e.g. criteria to include a given record or not, choice on the method used to derive the SST reconstruction and dating strategies, comparing model simulations with several syntheses relying on a coherent temporal framework has the added value to provide  a broader overview and different possible scenarios. As far as we know it is common practice to evaluate simulations or present-day climate to multiple observation datasets considered independently because of different blending methods and assimilation systems. This is a point that we mention in the revised version of the manuscript.

"It is indeed a common practice to evaluate model simulations against multiple observation datasets that are considered independent because of different compilation methods."

Hosing model simulation

The hosing experiment indeed reduced the RMSE between model and data. However, this improvement cannot be attributed to the 3,000-year hosing, without comparing 128

ka simulation without hosing with paleo data. And the length of the hosing period cannot be justified without comparing the 1,600-year hosing experiment (Holloway et al 2018 from the same group) to the same paleo data.

Response: We appreciate these points. We did compare a 128-ka simulation without hosing with the paleo data, which gives similar results as the 127-ka simulation. We mentioned it in the text as below.

"For comparison, we also run a standard 127-ka simulation of HadCM3 (HadCM3_127k), which gives qualitatively consistent results with a standard 128-ka simulation of HadCM3 by Holloway et al. (2018, not shown)."

We also compared the 2000-year hosing simulation with the paleo data, and it shows qualitatively consistent results with the 3000-year hosing simulation. This is because the climate tends to reach an equilibrium state after around 2000 years, as shown in Fig A1. However, we still consider the length of 3000 years to be valuable, as it provides a guidance for future studies on what are the optimal modelling length considering both computational costs and expected signals.

Additionally, I doubt if RMSE is suitable for evaluating model-data agreements. It appears that the larger RMSE is driven by the systematic offset between data and model. Investigating RMSE ignores spatial patterns of warming in different regions.

Response: We consider RMSE to be a suitable index to measure exactly "the systematic offset between data and model". It is true that RMSE cannot reveal spatial patterns, as it is mainly regional "average". We refer to the maps regarding spatial patterns of climate anomalies.

Some minor points

Line 76: more details are needed. How do these parameters differ from pi control?

Response: We added the following sentence:

The greenhouse gas concentrations were lower at 127 ka than preindustrial: 275 vs. 284.3 parts per million (ppm) for carbon dioxide, 685 vs. 808.2 parts per billion (ppb) for methane, and 255 vs. 273.0 ppb for nitrous oxide. At 127 ka, the Earth's orbit was characterised by a perihelion close to the boreal summer solstice, larger eccentricity, and higher obliquity than preindustrial (Berger and Loutre, 1991). Such configuration affected the seasonal and latitudinal distribution of solar insolation at the top of the atmosphere, resulting in a small positive annual insolation anomaly at 127 ka than preindustrial at high latitudes (Otto-Bliesner et al., 2017).

Line 79: why mention the CNRM model specifically here?

Response: Because this is the only model that does not use greenhouse gas concentrations at 127 ka for *lig127k*.

Line 89: Annual SST and summer SST in the paleo data syntheses are often derived from the same dataset but with different calibrations (e.g., alkenone, see Chandler and Langebroek 2021). Therefore, these two SSTs in paleo data sets are not independent. This point should be made in the methods.

Response: We appreciate this point and added the following sentence in Line 100:

"Note that annual and summer SST could be reconstructed from the same proxy using different calibration functions (Chandler and Langebroek, 2021b)."

Line 149: Forcing parameters for the 128 ka simulation need to be described in detail for comparison with the 127 ka simulation

Response: We added the following details in Line 167:

"The greenhouse gas concentrations in this simulation are close to those set by the PMIP4 lig127k guideline: carbon dioxide at 275 ppm, methane at 706.8 ppb, and nitrous oxide at 266 ppb. The vegetation, aerosol, and ice sheets were set identical to the corresponding preindustrial simulation (Tindall et al., 2009)."

Table 3: Good to add mean deference between pi control and HadSST1. From Fig. 2, the mean difference can be large for some models. How does this contribute to offset between the model (comparing 127 ka with pi) and data ( comparing 127 ka with HadISST1)?

Response: Thank you. We calculated mean differences between piControl and HadISST1, but we consider RMSE to be a better measure of model bias here, as mean differences are affected by the compensation between positive and negative bias. We added the following sentence in Line 191:

"We used RMSE to measure model-data discrepancies rather than mean differences to avoid compensation between positive and negative bias."

It is a very good question about how model bias in the preindustrial condition affects the simulated anomalies at 127 ka. While it is obvious that large warm bias in MIROC-ES2L undermines its applicability for a warmer climate (e.g. no sea ice to reduce), it is more complicated to draw any conclusions for other models. We also do not find a systematic relationship between the model bias and simulated anomalies. It is because of the model bias that we decided to focus on climate anomalies, rather than climate states at 127 ka.

Line 201: if you mean statistically significant, show the statistics.

Response: We did the student's t-test on the differences between annual SST in lig127k and piControl simulations in Fig. 3. Any differences we show in colour in Fig. 3 are statistically significant at 5% level.

Original text: "While the magnitude of simulated climate anomalies at 127 ka is small, the differences between lig127k and piControl are generally **significant** (Fig. 3)."

Line 215: How many is "a few"

Response: We modified this sentence:

"ACCESS-ESM1-5 and FGOALS-g3 show reduced September SIC over the southern Indian Ocean (Fig. 6a and 6f)."

---

## Author Response (AR2)

**Responses to Report #2**

I would like to thank the authors for their thoughtful replies and corresponding revisions following the first review. The points I raised have mostly been addressed. Just one point remains, which in my opinion was not adequately addressed and I really would like to see clarified. It concerns the decision to not use the Turney et al. 2020 SST synthesis, on the basis that it lacks a chronological framework and that it is a peak-warmth synthesis.

Response: We appreciate the efforts and time of Referee #2. We fully understand the concerns on the usage of the Turney et al. (2020) SST synthesis. Detailed arguments are presented in the following responses.

I agree that a peak-warmth synthesis is in general not appropriate for a model-data comparison, especially at global scale. However, was it not the case that Capron et al. 2014/2017 dated their Southern Ocean LIG SST records by aligning SST with the EDC temperature record? For a time slice close to the EDC temperature peak, the Capron method implicitly creates a peak-warmth synthesis because the SST temperature peaks are all aligned to the EDC temperature peak at ~128 ka. Capron et al. justified this methodology by assuming regionally synchronous SST changes through the LIG. Quoting from Capron et al. (2014):
"We follow the strategy of Govin et al. (2012) to align marine records onto the AICC2012 ice core chronology. It is based on the assumption that surface-water temperature changes in the sub-Antarctic zone of the Southern Ocean (respectively in the North Atlantic) occurred simultaneously with air temperature variations over inland Antarctica (respectively Greenland)."

Turney et al. 2020 use the original chronologies, which again is an approach I don't think is appropriate if using a global dataset, and I agree with the authors this is in some ways a "step back" in terms of progress addressing chronological frameworks etc (see the authors' reply to reviewer #1). But on the other hand, you would not be using all the Turney et al records, as you would only be using a Southern Ocean subset. When selecting only Southern Ocean records, the Turney et al method of finding peak warmth in the period 124-129 ka is conceptually very similar to Capron et al. 2017 in which Southern Ocean SST records are aligned by peak warmth, as a means of obtaining the 127ka SST. If Capron et al can argue that Southern Ocean SST peaks were synchronous, why reject that same argument from Turney et al?

In my view, for the 127 and 128 ka time slices that are so close to the Antarctic Ice Sheet LIG temperature peak, then either (i) both the Capron methodology and that of the Turney et al "peak warmth" are acceptable for comparison with model data in the Southern Ocean, or (ii)

neither are. Hence, the decision to use Capron et al. 2017 but not Turney et al. 2020, needs to be much stronger and not dependent on the argument against the peak warmth approach.

Response: Thank you for this thoughtful comment. Firstly, we would like to clarify that the dating strategy of the Capron et al. (2014, 2017) dataset is different from the peak warmth approach used in the Turney et al. (2020) dataset. It is true that the Capron et al. (2014, 2017) dataset align the Southern Ocean marine SST record onto the EDC water isotope profile to get the marine records on the AICC2012 timescales using the assumption the referee copied above. However, in practice they did not use the peak warmth to define a tie point for the climatic alignments. Instead, they used the mid-point during the warming phase over Termination 2 and the mid-point during the cooling over the glacial inception before the first millennial-scale climate variability (see Fig. 2 in Capron et al. 2014). This means that the Capron et al. (2014, 2017) dataset did not fix by construction the timing of the peak warmth in the Southern Ocean to the EDC peak warmth.

We do not think the Turney et al. (2020) dataset can be treated equally as the current four syntheses, because it completely ignores age scale issues (they take records on their published original chronologies) and does not provide age uncertainty estimates. It is true that the other datasets are associated with some limitations (e.g. based on climatic assumptions that are not fully satisfactory), but they made efforts in homogenising the chronologies onto a reference age scale and in quantifying the associated uncertainties. It is the way forward as clearly stated in many papers from the large paleoclimate communities over the past few years including the Otto-Bliesner et al. (2017) paper presenting the PMIP4 lig127k guidelines. Moreover, unlike the Capron et al. (2017) and Hoffman et al. (2017) datasets that provide 127 ka time slice reconstructions representing 126-128 ka with age and reconstruction uncertainties, the Turney et al. (2020) dataset provides peak reconstruction values during 129-124 ka. If we use the Turney et al. (2020) dataset to evaluate the model simulation of 127 ka climate, the potential age differences (among the records in the Turney et al. (2020) dataset and between the average age of the records in the Turney et al. (2020) dataset and the 127 ka) would prevent us from drawing any conclusions about model-data differences.

We add the following paragraph in Section 4.1 to reasoning our choice of not including this synthesis:

*"We note that a recent synthesis by Turney et al. (2020) compiles maximum annual SST estimates during the early LIG (129-124 ka), of which 28 records are located south of 40° S. The results from the model-data comparison using this recent dataset are similar to those obtained using the Capron et al. (2017) and Hoffman et al. (2017) syntheses. However, we do not include them in our study considering the strong limitations associated to the Turney et al. (2020) compilations related to the fact that the compiled records are kept on their original age scales and peak values are provided without quantitative age uncertainty estimates,. The*

*potential age differences among the records in the Turney et al. (2020) dataset, and between the average age of the records in the Turney et al. (2020) dataset and 127 ka would prevent us from drawing robust conclusions about model-data discrepancies."*

Minor points... line numbers from the tracked changes version.

New Fig 2 Taylor diagram: suggest to avoid green and red, for colourblind readers.
Response: Thank you. As mentioned, we only show the Taylor diagram as responses to comments, not in the manuscript.

L58 "while applying" change to "after applying"
Response: changed.

L88 grammar, "… resulting in a small positive annual insolation anomaly at 127 ka than preindustrial at high latitudes".
Response: changed to "…, which results in a small positive anomaly of annual insolation at 127 ka compared to preindustrial at high latitudes."

L94 grammar "…we use 100-year simulation from the end of each model integration period" change to "…we use 100-year simulations from the ends of each model integration period" or simply "we use the last 100 years of each simulation".
Response: changed to "we use 100-year simulations from the ends of each model integration period".

L383 Antarctic ice sheet should be in capitals (Antarctic Ice Sheet).
Response: changed.

L384 "may contribute to " change to "may have contributed to".
Response: changed.

**Responses to Report #3**

Gao et al improved the manuscript a lot in this revision. I am convinced of the main conclusion that freshwater input may be a key factor to simulate the warm conditions during the Last Interglacial. However, I think a few issues still need to be addressed.

Response: We are grateful for the comments of Referee #3, which help us improve the manuscript.

Data synthesis

I still believe and the authors probably agree that a better data synthesis would help with the model data comparison a lot. I also agree that publishing of the manuscript should not be prevented from not making a new data synthesis, if it is the convention in the field, as the authors explained. I do think, however, that the manuscript may be improved without too much effort in data compilation. Maybe it is good to at least get rid of some inconsistency between different compilations by getting averages for SSTs at the same sites.

Response: We fully agree that a better synthesis is extremely beneficial for model-data comparison, as discussed in Section 4.1. We understand it is not optimal to keep the inconsistency between different syntheses in the manuscript, but unfortunately, we are not in a position to favour one published SST reconstruction over another. The various methods used in each synthesis could explain some of the differences, and we believe more thorough work should be conducted to homogenise the SST reconstruction from proxies. We think simply averaging the reconstructions would mask the underlying issues potentially arising from different age scales, calibration functions, and reconstruction methods.

Null hypotheses

Following a point raised by Reviewer 2, the null hypothesis is not clearly described in the revision, and further clarification is needed.

I thought the null hypothesis presented by the author is comparing reconstructions at 127 ka with HadlSST (bottom line in Table 4), with the assumption that simulations by models are the same at 127 ka and pi condition. This point should be clearly mentioned somewhere in section 3.3. In this section, the authors compared the means between reconstructions with HadlSST, without setting the scene for comparing RMSEs against null hypotheses..

However, given the very large difference between your simulations and HadlSST, maybe an alternative set of null hypotheses is comparing reconstructions with simulations in both 127 ka and pi conditions? In this case, there are 12X4=48 null hypotheses associated with a different RMSE for SST. I am not saying this alternative is better, but the authors should better clarify the null hypothesis and consider this alternative way to set up the comparison. Moreover, it seems to me that when comparing RMSEs from model-data comparison with the null hypothesis, some kind of statistical test is needed. For example, looking at the

comparison between models and SST from EC2017 against the null hypothesis, I am not sure how confident the authors are in the statement that MIROC-ES2L (RMSE=4.1) outperforms the null hypotheses while ACCESS-ESM1-5 (RMSE=4.3) does not. If the test is similar to the t-test, then I think you can also take into consideration the number of observations in the reconstruction in assessing the robustness of the conclusion.

Response: Thank you. Firstly, we would like to clarify that we introduce a Null Scenario, not a hypothesis. In the Null Scenario, the 127 ka climate is assumed to be the same as the preindustrial climate, so the SST anomalies at each core site are zero in the Null Scenario. Then, when we compare the Null Scenario with a synthesis, e.g. annual SST from the Capron et al. (2017) dataset, we obtain a RMSE of 4.2°C. Intuitively, for a model simulation to be useful, when compared with a synthesis, it must demonstrate a lower RMSE than the Null Scenario.

The student t-test may not be applicable here. Firstly, it tests the significance of differences in mean values, not RMSE. Secondly, it normally requires row data with more than 30 entries, but we have only limited records for each dataset. We are not aware of any other suitable statistical tests, but we consider the comparison provides valuable qualitative insights.

We added the following sentence in line 242 to clarify on the Null Scenario: "Then when we compare the Null Scenario with a synthesis, e.g. annual SST reconstructions from the Capron et al. (2017) dataset, we obtain a RMSE of 4.2°C, which serves as a baseline to evaluate model performance"

Detailed comments

Line 144: mentioning HadlSST before introducing to readers what this is.

Response: We introduce HadISST1 now at line 114 when we first mention it: "the HadISST1 dataset (1870-1899), which contains global monthly SST and SIC on 1°×1° grids from 1870 to present and is constructed by the UK Met Office using multiple observational data sources (Rayner et al., 2003)."

Section 2.4 Not sure why you compare model results with SST from the European Space Agency Climate Change Initiative rather than HadlSST during the same period?

Response: Thank you for this comment. The SST dataset from the ESA CCI project has much finer temporal (daily) and spatial (0.05 degree) resolution. Although the spatiotemporal resolution of HadISST1 is already enough for model evaluation, we consider checking different datasets to be a good way to indicate the robustness of our results.

We modified the following sentence in line 183: *"Indeed, we also compared annual SST between 1982-2014 in CMIP6 historical simulations of the 12 models with a SST dataset*

*from the European Space Agency Climate Change Initiative (Merchant et al., 2014), **which has much finer spatial and temporal resolution than HadISST1**."*